# The inherent mutational tolerance and antigenic evolvability of influenza hemagglutinin

**Bargavi Thyagarajan[1,2], Jesse D Bloom[1,2]\***

[1]Division of Basic Sciences, Fred Hutchinson Cancer Research Center, Seattle, United States; [2]Computational Biology Program, Fred Hutchinson Cancer Research Center, Seattle, United States

**Abstract** Influenza is notable for its evolutionary capacity to escape immunity targeting the viral hemagglutinin. We used deep mutational scanning to examine the extent to which a high inherent mutational tolerance contributes to this antigenic evolvability. We created mutant viruses that incorporate most of the $\approx 10^4$ amino-acid mutations to hemagglutinin from A/WSN/1933 (H1N1) influenza. After passaging these viruses in tissue culture to select for functional variants, we used deep sequencing to quantify mutation frequencies before and after selection. These data enable us to infer the preference for each amino acid at each site in hemagglutinin. These inferences are consistent with existing knowledge about the protein's structure and function, and can be used to create a model that describes hemagglutinin's evolution far better than existing phylogenetic models. We show that hemagglutinin has a high inherent tolerance for mutations at antigenic sites, suggesting that this is one factor contributing to influenza's antigenic evolution.

## Introduction

Epidemic influenza poses an annual threat to human health largely because the virus rapidly evolves to escape the immunity elicited by previous infections or vaccinations. The most potent form of anti-influenza immunity is antibodies targeting the virus's hemagglutinin (HA) protein (*Yewdell et al., 1979*; *Wiley et al., 1981*; *Caton et al., 1982*). The virus evades these antibodies primarily by accumulating amino-acid substitutions in HA's antigenic sites (*Smith et al., 2004*; *Das et al., 2013*; *Koel et al., 2013*; *Bedford et al., 2014*). Remarkably, HA undergoes this rapid evolution while retaining the ability to fold to a highly conserved structure that performs two functions essential for viral replication: receptor binding and membrane fusion (*Wiley and Skehel, 1987*; *Russell et al., 2004*). HA is therefore highly 'antigenically evolvable' in the sense that it can accommodate rapid antigenic change without compromising its structural and functional properties.

Two factors that undoubtedly contribute to HA's rapid antigenic evolution are influenza's high mutation rate and the strong selection that immunity exerts on the virus. However, it is unclear whether these factors are sufficient to fully explain HA's antigenic evolution. For instance, while some other error-prone viruses (such as HIV and hepatitis C) also exhibit rapid antigenic evolution of their surface proteins (*Burton et al., 2012*), other viruses with comparable mutation rates (such as measles) show little propensity for antigenic change (*Sheshberadaran et al., 1983*; *Duffy et al., 2008*), despite the fact that evasion of immunity would presumably confer a selective benefit. A variety of explanations ranging in scale from ecological to molecular can be posited to account for these differences in rates of antigenic evolution (*Lipsitch and O'Hagan, 2007*; *Koelle et al., 2006*; *Heaton et al., 2013*). One hypothesis is that HA has a high inherent tolerance for mutations in its antigenic sites, thereby conferring on influenza the evolutionary capacity to escape from anti-HA antibodies with relative ease.

**\*For correspondence:** jbloom@fhcrc.org

**Competing interests:** The authors declare that no competing interests exist.

**eLife digest** Influenza is a major threat to human health largely because the flu virus evolves rapidly to escape recognition by the immune system. These ongoing changes also explain why flu vaccines become less effective over time and need to be reformulated every year.

Hemagglutinin is a protein on the surface of the flu virus that helps the virus bind to and infect host cells. The surface proteins of most viruses are recognized by the immune system, and influenza hemagglutinin is no exception. However, hemagglutinin is unusual in that it evolves exceptionally rapidly to avoid being recognized by the immune system. This raises an important question: what is it about the influenza hemagglutinin protein that allows it to change so readily?

Thyagarajan and Bloom address this question by making mutant copies of the gene that encodes the hemagglutinin protein. There are over 10,000 ways in which the protein can be mutated, and Thyagarajan and Bloom managed to make the vast majority of the possible changes. The mutated genes were then re-introduced into the virus, and the mutant viruses were allowed to replicate in cells for several generations.

Thyagarajan and Bloom sequenced the viruses that had replicated—which meant that the mutant copies of the hemagglutinin protein in these viruses still worked—and looked to see where in the protein the changes had occurred. Those regions that rarely changed included the part of the protein that binds to host cells, whereas other regions—especially those that are recognized by the immune system—were much more likely to contain mutations. Thyagarajan and Bloom then went on to show that not all influenza proteins share hemaglutinin's capacity to change the regions targeted by the immune system, suggesting that this capacity is possibly a unique feature of this protein.

Thyagarajan and Bloom also suggest that this capacity to tolerate mutations in parts of proteins that are recognized by the immune system might be important for shaping a virus's ability to evolve to escape this recognition. Future work is now needed to see how tolerant to mutations other viral proteins are, and to reveal which properties of a protein determine its tolerance to mutations.

Testing this hypothesis requires quantifying the inherent mutational tolerance of each site in HA. This cannot be done simply by examining variability among naturally occurring viruses, since the evolution of influenza in nature is shaped by a combination of inherent mutational tolerance and external immune selection. For example, the rapid evolution of HA's antigenic sites in nature could reflect the fact that these sites are especially tolerant of mutations, or it could be purely a consequence of strong immune selection. Traditional experimental approaches using site-directed mutagenesis or serial viral passage are also inadequate to quantify inherent mutational tolerance—while such experimental techniques have been used to determine the effect of specific mutations on HA, they cannot feasibly be applied to all possible individual amino-acid mutations. Recently *Heaton et al. (2013)* used transposon mutagenesis to show that HA is tolerant to the random insertion of five to six amino-acid sequences at several locations in the protein. However, the relevance of this tolerance to insertional mutations is unclear, since HA's actual antigenic evolution involves almost entirely point substitutions, with only a very low rate of insertions and deletions.

Here we use the new high-throughput experimental technique of deep mutational scanning (*Fowler et al., 2010*; *Araya and Fowler, 2011*) to comprehensively quantify the tolerance of HA to amino-acid mutations. Specifically, we create mutant libraries of the HA gene from the H1N1 strain A/WSN/1933 (WSN) that contain virtually all of the $\approx 4 \times 10^4$ possible individual codon mutations, and therefore virtually all of the $\approx 10^4$ possible amino-acid mutations. We use these mutant libraries to generate pools of mutant influenza viruses, which we estimate incorporate at least 85% of the possible HA codon mutations and 97% of the possible amino-acid mutations. We then passage these viruses to select for functional variants, and use Illumina deep sequencing to determine the frequency of each HA mutation before and after this selection for viral growth. Since these experiments measure the impact of mutations in the absence of immune selection, they enable us to quantify HA's inherent preference for each amino acid at each site in the protein. We show that these quantitative measurements are consistent with existing knowledge about HA structure and function, and can be used to create an

evolutionary model that describes HA's natural evolution far better than existing models of sequence evolution. Finally, we use our results to show that HA's antigenic sites are disproportionately tolerant of mutations, suggesting that a high inherent tolerance for mutations at key positions targeted by the immune system is one factor that contributes to influenza's antigenic evolvability.

# Results

## Strategy for deep mutational scanning of HA

Our strategy for deep mutational scanning (*Fowler et al., 2010*; *Araya and Fowler, 2011*) of HA is outlined in *Figure 1*. The wildtype WSN HA gene was mutagenized to create a diverse library of mutant HA genes. This library of mutant genes was then used to generate a pool of mutant viruses by reverse genetics (*Hoffmann et al., 2000*). The mutant viruses were passaged at a low multiplicity of infection to ensure a linkage between genotype and phenotype. The frequencies of mutations before and after selection for viral growth were quantified by Illumina deep sequencing of the mutant genes (the **mutDNA** sample in *Figure 1*) and the mutant viruses (the **mutvirus** sample in *Figure 1*). An identical process was performed in parallel using the unmutated wildtype HA gene to generate unmutated viruses in order to quantify the error rates associated with sequencing, reverse transcription, and virus growth (these are the **DNA** and **virus** samples in *Figure 1*). The entire process in *Figure 1* was performed in full biological triplicate (the replicates are referred to as #1, #2, and #3). In addition, a repeat of the Illumina sample preparation and deep sequencing was performed for replicate #1 to quantify the technical variation associated with these processes.

## Creation of HA codon-mutant libraries

The deep mutational scanning strategy in *Figure 1* requires creating mutant libraries of the HA gene. We wanted to assess the impact of all possible amino-acid mutations. Most mutagenesis techniques operate at the nucleotide level, and so frequently introduce single-nucleotide codon changes (e.g., GGA → cGA) but only very rarely introduce multi-nucleotide codon changes (e.g., GGA → cat). However, several PCR-based techniques have recently been developed to introduce random codon mutations into full-length genes (*Firnberg and Ostermeier, 2012*; *Bloom, 2014*; *Jain and Varadarajan, 2014*). We used one of these techniques (*Bloom, 2014*) to create three replicate codon-mutant libraries of the WSN HA gene (*Supplementary file 1*).

Sanger sequencing of 34 individual clones indicated that the libraries contained an average of slightly over two codon mutations per gene, with a very low rate of insertions and deletions (less than 0.1 per gene). The number of mutations per clone was distributed around this average in an approximately Poisson fashion (*Figure 2*). The mutations consisted of a mix of one-, two-, and three-nucleotide codon changes, and were roughly uniform in their nucleotide composition and location in the gene (*Figure 2*).

The genes in each mutant library were cloned at high efficiency into a bidirectional influenza reverse-genetics plasmid (*Hoffmann et al., 2000*). Each of the library replicates contained at least six-million unique clones—a diversity that far exceeds the $10^4$ unique single amino-acid mutations and the $\approx 4 \times 10^4$ unique single codon mutations to the HA gene. The vast majority of possible codon and amino-acid mutations are therefore represented many times in each plasmid mutant library, both individually and in combination with other mutations.

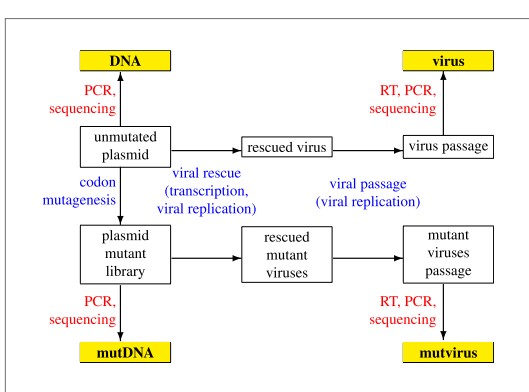

**Figure 1**. Schematic of the deep mutational scanning experiment. The Illumina deep-sequencing samples are shown in yellow boxes (**DNA**, **mutDNA**, **virus**, **mutvirus**). Experimental steps and associated sources of mutations are shown in blue text, while sources of error during Illumina sample preparation and sequencing are shown in red text. This entire process was performed in biological triplicate.

## Generation of mutant viruses by reverse genetics

The HA plasmid mutant libraries were used to generate pools of mutant influenza viruses by reverse genetics (*Hoffmann et al., 2000*). Briefly, this

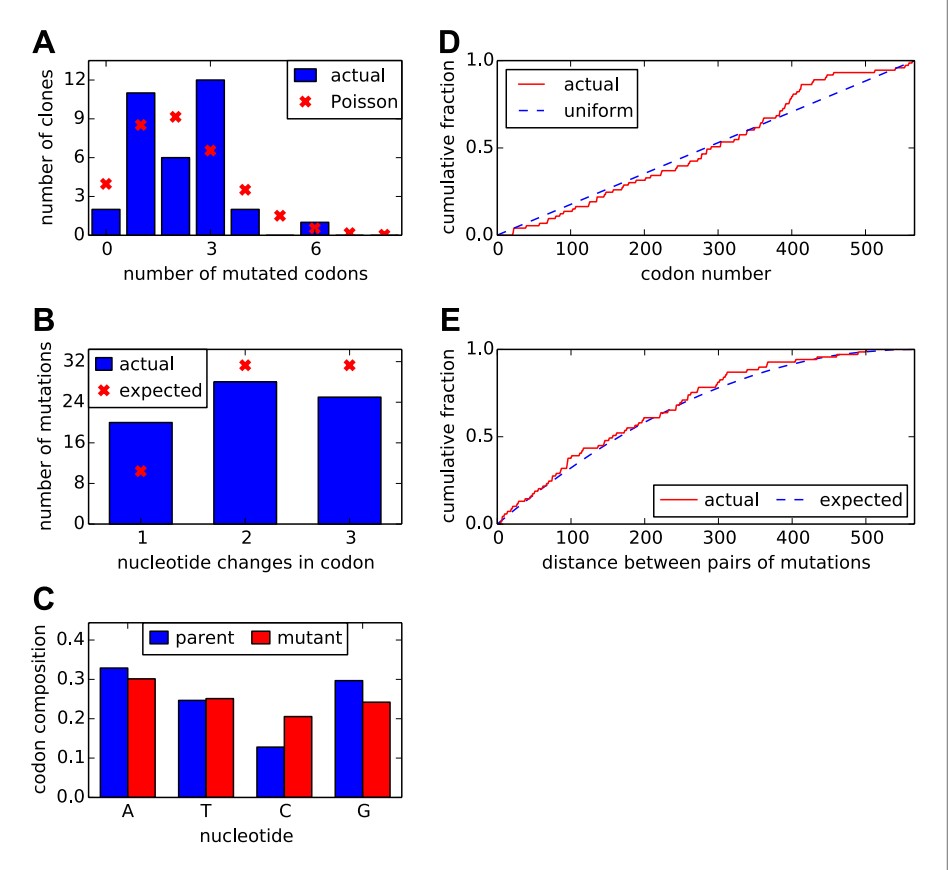

**Figure 2**. Properties of the HA codon-mutant library as assessed by Sanger sequencing of 34 individual clones drawn roughly evenly from the three experimental replicates. (**A**) There are an average of 2.1 codon mutations per clone, with the number per clone following a roughly Poisson distribution. (**B**) The codon mutations involve a mix of one-, two-, and three-nucleotide mutations. (**C**) The nucleotide composition of the mutant codons is roughly uniform. (**D**) The mutations are distributed uniformly along HA's primary sequence. (**E**) There is no tendency for mutations to cluster in primary sequence. Shown is distribution of observed pairwise distances between mutations in multiply mutated clones vs the expected distribution when the mutations are placed independently in the clones. All plots show results only for substitution mutations; insertion/deletion mutations are not shown. However, only two insertion/deletion mutations (0.06 per clone) were identified. The data and computer code used to generate this figure are at https://github.com/jbloom/SangerMutantLibraryAnalysis/tree/v0.2.

process involves transfecting cells with the HA plasmid mutant library along with plasmids encoding the other seven genes from the WSN strain of influenza. Although the cells were transfected with a very large diversity of HA plasmids, we were uncertain what fraction of the genes encoded on these plasmids would actually be productively packaged into a virus. In an attempt to maximize the diversity in the viral pools, the mutant viruses were generated by transfecting several dozen wells of cells. The logic behind this scheme was to maintain substantial diversity even if only a subset of viral mutants stochastically predominated in each individual well of cells. A different replicate virus pool was generated for each of the three HA plasmid mutant libraries.

The mutant viruses generated for each replicate were passaged at a relatively low multiplicity of infection (MOI) of 0.1 to reduce the probability of co-infection, thereby creating a link between viral genotype and phenotype. This genotype-phenotype link is essential to ensure that the sequenced HA gene matches the protein on the surface of the virus. Our previous work with NP (**Bloom, 2014**) indicates that one low MOI passage is sufficient to create a strong genotype-phenotype link, since that previous work found that the results obtained after one viral passage were extremely similar to the results obtained after two viral passages. In order to maintain a diversity of over two-million infectious viral particles, we performed the passaging in a total of $2.4 \times 10^7$ cells.

## Deep sequencing reveals purifying selection against many mutations

We used Illumina sequencing to quantify the frequencies of mutations before and after selection for viral growth. For each replicate, we sequenced HA from the unmutated plasmid, the plasmid mutant library, virus produced from the unmutated plasmid, and mutant virus produced from the plasmid mutant library—these are the **DNA**, **mutDNA**, **virus**, and **mutvirus** samples in *Figure 1*. For the **DNA** and **mutDNA** samples, the HA gene was amplified directly from the plasmids by PCR. For the **virus** and **mutvirus** samples, the HA gene was first reverse-transcribed from viral RNA and was then amplified by PCR. In all cases, template quantification was performed prior to PCR to ensure that >$10^6$ initial HA molecules were used as templates for subsequent amplification.

In order to reduce the sequencing error rate, the HA molecules were fragmented to roughly 50 nucleotide fragments using Illumina's transposon-based Nextera kit, and then sequenced with overlapping paired-end reads (*Figure 3—figure supplement 1*). We only called codon identities for which both paired reads concur—this strategy substantially increases the sequencing fidelity, since it is rare for the same sequencing error to occur in both reads. For each sample, we obtained in excess of $10^7$ overlapping paired-end reads that could be aligned to HA (*Figure 3—figure supplement 2*). As shown in *Figure 3—figure supplement 3*, the read depth varied somewhat along the primary sequence, presumably due to known weak biases in the insertion sites for the Nextera transposon (*Adey et al., 2010*). However, these biases were fairly mild, and so we obtained well over $2 \times 10^5$ unique paired reads for nearly all HA codons.

*Figure 3* shows the frequency of mutations in each sample as quantified by deep sequencing. The **DNA** samples derived from unmutated HA plasmid show a low frequency of apparent mutations which are almost exclusively composed of single-nucleotide codon changes—the frequency of these apparent mutations reflects the rate of errors from the PCR amplification and subsequent deep sequencing. The **virus** samples created from the unmutated plasmid show only a slightly higher frequency of mutations, indicating that reverse-transcription and viral replication introduce only a small number of additional mutations. As expected, the **mutDNA** samples derived from the plasmid mutant libraries show a high rate of one-, two-, and three-nucleotide mutations, as all three types of mutations were introduced during the codon mutagenesis. The **mutvirus** samples derived from the mutant virus pools exhibit a mutation rate that is substantially lower than that of the **mutDNA** samples. Most of the reduction in mutation frequency in the **mutvirus** samples is due to decreased frequencies of nonsynonymous and stop-codon mutations; synonymous mutations are only slightly depressed in frequency. As stop-codon and nonsynonymous mutations are much more likely than synonymous mutations to substantially impair viral fitness, these results are consistent with purifying selection purging deleterious mutations during viral growth.

Inspection of *Figure 3* also demonstrates an important advantage of introducing the mutations at the codon rather than the nucleotide level. While there is a low but non-zero rate of errors (from sequencing, PCR, or reverse-transcription) that lead to single-nucleotide codon changes (as judged by the **DNA** and **virus** samples), errors that lead to multi-nucleotide codon changes are negligible because it is extremely rare for a single codon to experience two errors. We similarly expect that any *do novo* mutations or reversions that arise during viral growth should be limited to single-nucleotide changes given the short duration of viral passage in our experiments. The fact that our mutant libraries were constructed at the codon rather than the nucleotide level means that the vast majority (54 of 63) possible mutations to each codon involve multiple nucleotide changes, and so the sequencing results for these mutations can be analyzed essentially at face value, without having to worry about confounding errors. For the remaining (9 of 63) possible mutations that only involve a single-nucleotide codon change, we have attempted to statistically correct for the error rates estimated from our controls as described in the 'Materials and methods'.

## Most mutations are sampled by the experiments

It is important to assess the completeness with which the experiments sampled all possible HA mutations. Several problems could limit mutational sampling: mutations might be absent from the plasmid mutant libraries due to biases in the codon mutagenesis, mutations that are present in the plasmid mutants might fail to be incorporated into viruses due to stochastic bottlenecks during virus generation by reverse genetics, or the sequencing read depth might be inadequate to sample the mutations that are present. The most straightforward way to assess these issues is to quantify the number of times that each possible multi-nucleotide codon mutation is observed in the **mutDNA**

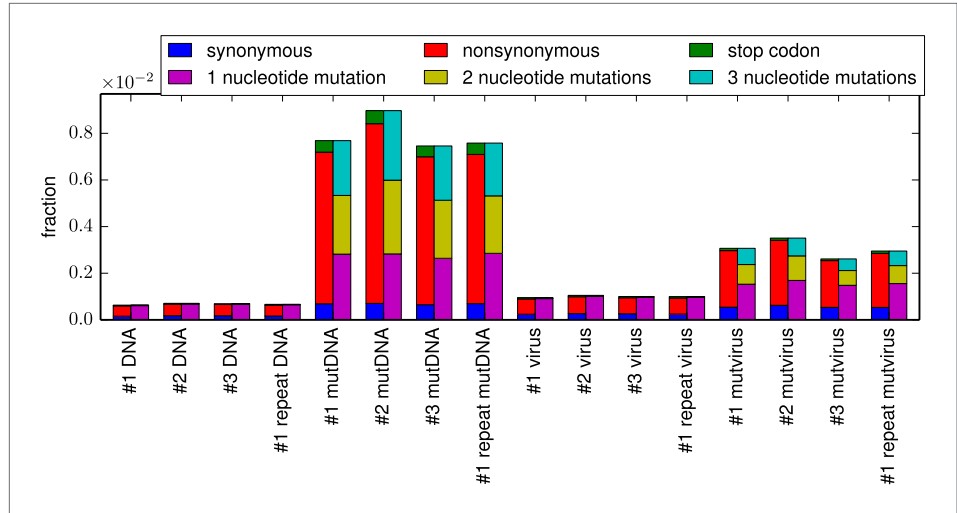

**Figure 3**. The per-codon frequencies of mutations in the samples. The samples are named as in *Figure 1*, with the experimental replicate indicated with the numeric label. The **DNA** samples have a low frequency of mutations, and these mutations are composed almost entirely of single-nucleotide codon changes—these samples quantify the baseline error rate from PCR and deep sequencing. The mutation frequency is only slightly elevated in **virus** samples, indicating that viral replication and reverse transcription introduce only a small number of additional mutations. The **mutDNA** samples have a high frequency of single- and multi-nucleotide codon mutations, as expected from the codon mutagenesis procedure. The **mutvirus** samples have a lower mutation frequency, with most of the reduction due to fewer stop-codon and nonsynonymous mutations—consistent with purifying selection purging deleterious mutations. The data and code used to create this plot is available via http://jbloom.github.io/mapmuts/example_WSN_HA_2014Analysis.html; this plot is the file *parsesummary_codon_types_and_nmuts.pdf* described therein. The sequencing accuracy was increased by using overlapping paired-end reads as illustrated in *Figure 3—figure supplement 1*. The overall number of overlapping paired-end reads for each sample is shown in *Figure 3—figure supplement 2*. A representative plot of the read depth across the primary sequence is shown in *Figure 3—figure supplement 3*.

The following figure supplements are available for figure 3:

**Figure supplement 1**. The overlapping paired-end Illumina sequencing strategy.

**Figure supplement 2**. The total number of reads for each sample.

**Figure supplement 3**. The per-codon read depth as a function of primary sequence.

and **mutvirus** samples. Restricting the analysis to multi-nucleotide codon mutations avoids the confounding effects of sequencing and reverse-transcription errors, which cause almost exclusively single-nucleotide changes.

*Figure 4* shows the number of times that each mutation was observed in the combined sequencing data for the three biological replicates; *Figure 4—figure supplement 1* shows the same data for the replicates individually. More than 99.5% of multi-nucleotide codon mutations are observed at least five times in the combined sequencing data from the plasmid mutant libraries (**mutDNA** samples), and ≈ 97.5% of all such mutations are observed at least five times in sequencing of the **mutDNA** for each individual replicate. These results indicate that the vast majority of codon mutations are represented in the plasmid mutant libraries.

In contrast, only 53% of multi-nucleotide codon mutations are observed at least five times in the combined sequencing data for the mutant viruses (**mutvirus** samples), and only ≈ 26% of such mutations are observed at least five times in sequencing of the **mutvirus** for each individual replicate (*Figure 4*, *Figure 4—figure supplement 1*). However, these numbers are confounded by the fact that many mutations are deleterious, and so may be absent because purifying selection has purged them from the mutant viruses. A less confounded measure is the frequency of *synonymous* multi-nucleotide mutations, since synonymous mutations are less likely to be strongly deleterious.

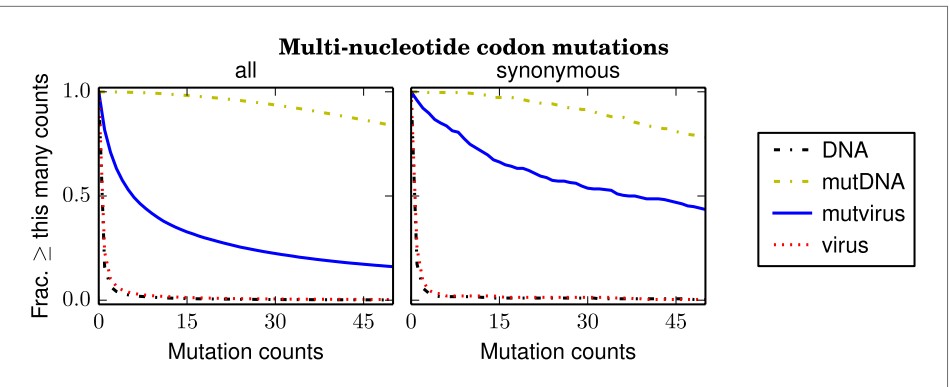

**Figure 4**. The number of times that each possible multi-nucleotide codon mutation was observed in each sample after combining the data for the three biological replicates. Nearly all mutations were observed many times in the **mutDNA** samples, indicating that the codon mutagenesis was comprehensive. Only about half of the mutations were observed at least five times in the **mutvirus** samples, indicating either a bottleneck during virus generation or purifying selection against many of the mutations. If the analysis is restricted to synonymous multi-nucleotide codon mutations, then about 85% of mutations are observed at least five times in the **mutvirus** samples. Since synonymous mutations are less likely to be eliminated by purifying selection, this latter number provides a lower bound on the fraction of codon mutations that were sampled by the mutant viruses. The redundancy of the genetic code means that the fraction of amino-acid mutations sampled is higher. The data and code used to create this figure are available via http://jbloom.github.io/mapmuts/example_WSN_HA_2014Analysis.html; this plot is the file *countparsedmuts_multi-nt-codonmutcounts.pdf* described therein. Similar plots for the individual replicates are shown in *Figure 4—figure supplement 1*.

The following figure supplements are available for figure 4:

**Figure supplement 1**. Plots like those in *Figure 4* for the individual biological replicates.

About 85% of such mutations are observed at least five times in the combined **mutvirus** samples, and ≈ 51% of such mutations are observed at least five times in the **mutvirus** samples for the individual replicates (*Figure 4*, *Figure 4—figure supplement 1*). Note that these numbers are only a lower bound on the fraction of codon mutations sampled by the mutant viruses—even synonymous mutations to influenza are sometimes strongly deleterious (*Marsh et al., 2008*), and so some of the missing synonymous codon mutations may have been introduced into mutant viruses but then purged by purifying selection. Furthermore, the redundancy of the genetic code means that the fraction of possible amino-acid mutations sampled is substantially higher than the fraction of codon mutations sampled. Specifically, if 85% of possible codon mutations are sampled at least five times in the combined libraries (as *Figure 4* indicates), then our simulations suggest that ≈ 97% of possible amino-acid mutations will have also been sampled at least five times ('Materials and methods').

Overall, these results indicate that nearly all mutations are represented in the plasmid mutant libraries. Virus generation by reverse genetics does introduce a bottleneck—but fortunately, this bottleneck is sufficiently mild that at least half of all possible codon mutations are still sampled at least five times by the mutant viruses in each individual replicate. Combining the data for the three replicates brings the coverage of possible codon mutations to around 85%, and the coverage of possible amino-acid mutations to 97%. Therefore, the sampling of mutations is sufficiently complete to provide information on the effects of most amino-acid mutations when the data from the three experimental replicates are combined.

## Estimation of the effects of each amino-acid mutation to HA

We quantified the effects of mutations in terms of site-specific amino-acid 'preferences'. These preferences are the expected frequency of each amino acid at each site in the mutant viruses in a hypothetical situation in which all amino acids are introduced at that site at equal frequency in the initial plasmid mutant library (*Bloom, 2014*). Because many of the HAs in our libraries contain several mutations, these preferences do not simply correspond to the fitness effect of each individual mutation to the WSN HA—rather, they represent the average effect of each mutation in a collection of closely related

HA mutants. Mutations to amino acids with high preferences are favored by selection, while mutations to amino acids with low preferences are disfavored. The amino-acid preferences are inferred from the deep sequencing data using a Bayesian statistical framework in which the observed counts are treated as draws from multinomial distributions with unknown parameters representing the initial mutagenesis rate, the various error rates, and selection as represented by the preferences (see 'Materials and methods' for details).

*Figure 5* shows the amino-acid preferences for the entire HA gene inferred from the combined data from all three biological replicates. As can be seen from this figure, some sites have strong preferences for one specific amino acid, while other sites are tolerant of a variety of different amino acids. As described in *Table 1*, the inferred amino-acid preferences are consistent with existing knowledge about mutations and residues affecting HA stability, membrane fusion, proteolytic activation, and receptor binding (*Nakajima et al., 1986*; *Martin et al., 1998*; *Qiao et al., 1999*; *Stech et al., 2005*). This concordance suggests that the deep mutational scanning effectively captures many of the structural and functional constraints on HA.

Despite the general concordance between the inferred amino-acid preferences and existing knowledge, it is important to quantify the experimental error associated with the deep mutational scanning. We sought to quantify two factors: *technical* variation due to inaccuracies and statistical limitations during Illumina sample preparation and deep sequencing, and *biological* variation due to stochasticity in the viral mutants that were generated and enriched during each replicate of the experiment. *Figure 6A* shows the correlation between biological replicate #1 and a technical repeat of the Illumina sample preparation and deep sequencing for this biological replicate. There is a very high correlation between the preferences inferred from these two repeats, indicating that technical variation has only a very minor influence on the final inferred amino-acid preferences. *Figure 6B–D* show the correlation among the three different biological replicates. Although the biological replicates are substantially correlated, there is also clear variation. Most of this variation is attributable to amino acids which in one replicate are inferred to have preferences near the a priori expectation of 0.05 (there are 20 amino acids, which in the absence of data are all initially assumed to have an equal preference of $\frac{1}{20}$), but in another replicate are inferred to have a much higher or lower preference. Such variation arises because the mutant viruses for each biological replicate only sample about 50% of the possible codon mutations (see previous section), meaning that there is little data for some mutations in any given replicate. Fortunately, combining the three biological replicates greatly increases the coverage of possible mutations (see previous section). Therefore, inferences made from the combined data (as in *Figure 5*) should be substantially more accurate than inferences from any of the individual replicates. This idea is supported by the results below, which quantify the extent to which the inferred preferences accurately describe natural HA evolution.

## Comparison to another high-throughput study of mutations to HA

As our paper was under review, *Wu et al. (2014)* published the results of using a similar strategy to examine the effects of mutations to the WSN HA. In their study, the HA gene was mutated at the nucleotide level, so their experiments surveyed only amino-acid mutations accessible by single-nucleotide codon changes. As a result, they provide data on the effects of only about 20% of the $19 \times 564 = 10716$ amino-acid mutations examined in our study. Despite this limitation, their study provides a large dataset of mutational effects to which we can compare our results.

*Figure 7* compares the mutational effects determined in our study to those from *Wu et al. (2014)*. There is a highly significant correlation between the results of the two studies—but the inferred mutational effects are certainly not identical. Because *Wu et al. (2014)* do not provide the data for replicates of their experiment, we are unable to assess whether the variability between the two different studies exceeds the variability between experimental replicates within each study. So one can imagine both biologically interesting and uninteresting explanations for the imperfect correlation between the results of the two studies. The interesting explanation is that differences in experimental methodology could lead to different selection pressures on specific mutations: for instance, *Wu et al. (2014)* use A549 cells while we use MDCK-SIAT1 cells, and perhaps the impact of certain mutations is dependent on the cell line. The uninteresting explanation is that the imperfect correlation is simply due to noise in the experimental measurements. Unfortunately, it is not straightforward to distinguish between these two explanations. This difficulty in pinpointing reasons

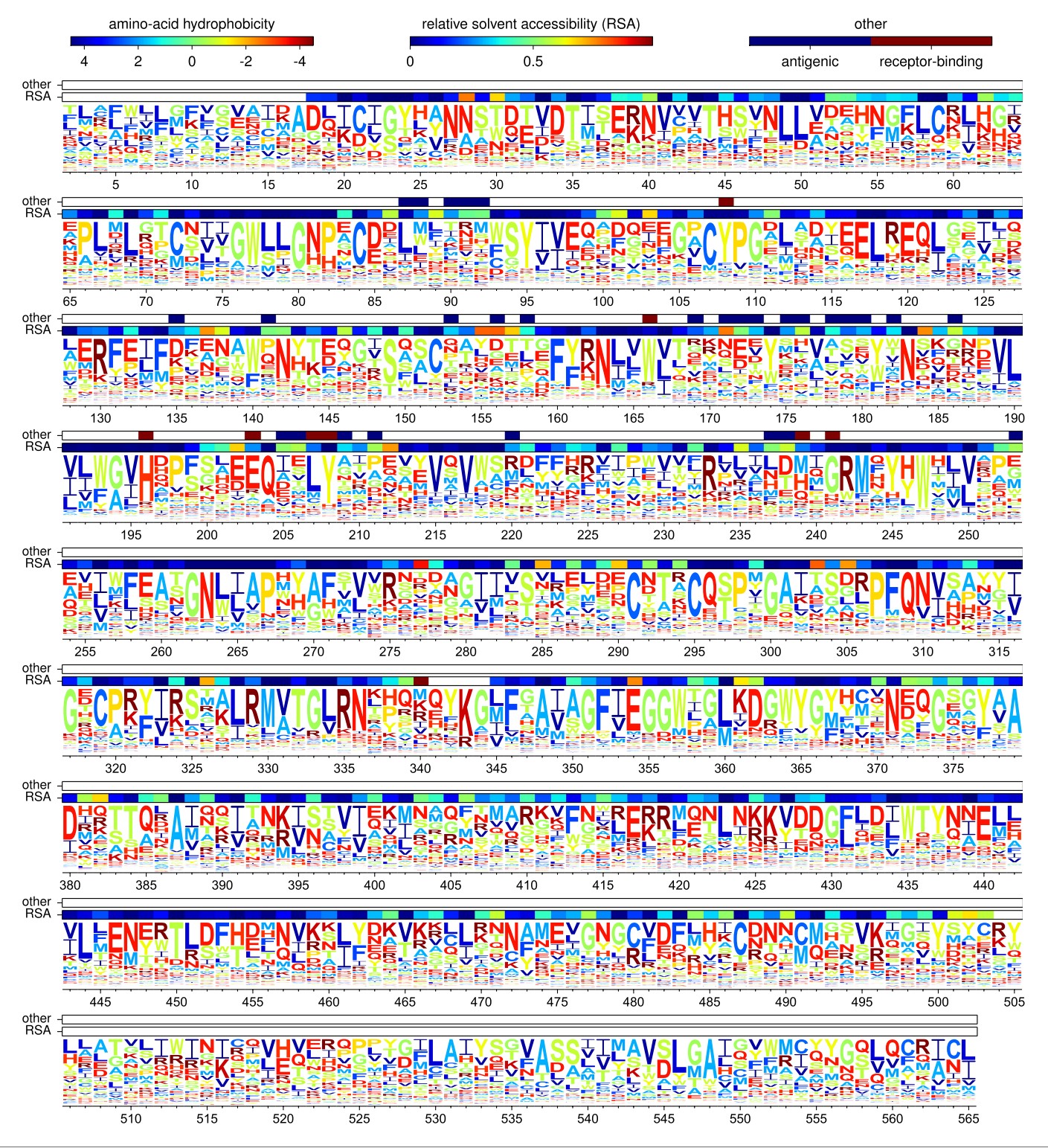

**Figure 5**. The amino-acid preferences inferred using the combined data from the three biological replicates. The letters have heights proportional to the preference for that amino acid, and are colored by hydrophobicity. The first overlay bar shows the relative solvent accessibility (RSA) for residues in the HA crystal structure. The second overlay bar indicates Caton et al. antigenic sites or conserved receptor-binding residues. The sequence is numbered sequentially beginning with 1 at the N-terminal methionine—however, this first methionine is not shown as it was not mutagenized. *Figure 5—figure supplement 1* shows *Figure 5. Continued on next page*

*Figure 5. Continued*
the same data with H3 numbering of the sequence. The data and code used to create this figure are available via http://jbloom.github.io/mapmuts/
example_WSN_HA_2014Analysis.html; this plot is the file *sequentialnumbering_site_preferences_logoplot.pdf* described therein.
The following figure supplements are available for figure 5:

**Figure supplement 1**. A plot matching that shown in **Figure 5** except that the HA sequence is numbered using the H3 numbering scheme.

for inter-study variation highlights a limitation of the high-throughput experimental methodology employed by ourselves and **Wu et al. (2014)**: while such experiments provide a wealth of data, numerous factors can create noise in these data (sequencing errors, population bottlenecks, epistasis among mutations, *etc*). Realizing the full potential of such studies will therefore require extensive experimental controls and biological replicates to quantify errors and noise to enable comparisons across data sets.

Nonetheless, **Figure 7** shows that there is a highly significant correlation between the results of these two high-throughput studies, despite differences in experimental methodology and unquantified sources of experimental noise. This fact suggests that both studies capture fundamental constraints on HA's mutational tolerance. In the remaining sections, we apply the more comprehensive data generated by our study to address questions about HA's natural evolution and antigenic evolvability.

**Table 1.** The amino-acid preferences inferred from the combined experimental replicates are consistent with existing knowledge about HA structure and function

| Site in sequential numbering | Site in H3 numbering | Existing knowledge | Inferred amino-acid preferences |
|---|---|---|---|
| 127 | 117 (HA1) | Mutation from S to P creates a temperature-sensitive defect (**Nakajima et al., 1986**) | The preference for S is 30 times higher than the preference for P |
| 174 | 161 (HA1) | Mutation from Y to H creates a temperature-sensitive defect (**Nakajima et al., 1986**) | The preference for Y is 25 times higher than the preference for H |
| 344 | 1 (HA2) | Mutation from G to E abolishes HA fusion activity (**Qiao et al., 1999**) | The preference for G is 11 times higher than for E |
| 343 | 327 (HA1) | A basic residue (R or K) is required for HA proteolytic activation (**Stech et al., 2005**) | The combined preferences for R and K (0.87) far exceed those of all other amino acids combined |
| 108 | 98 (HA1) | Receptor-binding residue, is Y in >99% of natural H1 HAs | The preference for Y (0.61) exceeds those of all other amino acids combined |
| 166 | 153 (HA1) | Receptor-binding residue, is W in >99% of natural H1 HAs | The preference for W (0.65) exceeds those of all other amino acids combined |
| 196 | 183 (HA1) | Receptor-binding residue, is H in >99% of natural H1 HAs | The preference for H (0.69) exceeds those of all other amino acids combined |
| 203 | 190 (HA1) | Receptor-binding residue, is D in 90% of natural H1 HAs | The highest preference is for the chemically similar E |
| 207 | 194 (HA1) | Receptor-binding residue, is L in 97% of natural H1 HAs | The preference for L (0.55) exceeds those of all other amino acids combined |
| 208 | 195 (HA1) | Receptor-binding residue, is Y in >99% of natural H1 HAs | The preference for Y (0.72) exceeds those of all other amino acids combined |
| 239 | 226 (HA1) | Receptor-binding residue, is Q in ≈99% of natural H1 HAs | Q is one of three amino acids with a high preference |
| 241 | 228 (HA1) | Receptor-binding residue, is G in >99% of natural H1 HAs | The preference for G (0.57) exceeds those of all other amino acids combined |

The conserved receptor-binding residues listed in this table are those delineated in the first table of **Martin et al. (1998)** that also have at least 90% conservation among all naturally occurring H1 HAs in the Influenza Virus Resource (**Bao et al., 2008**).

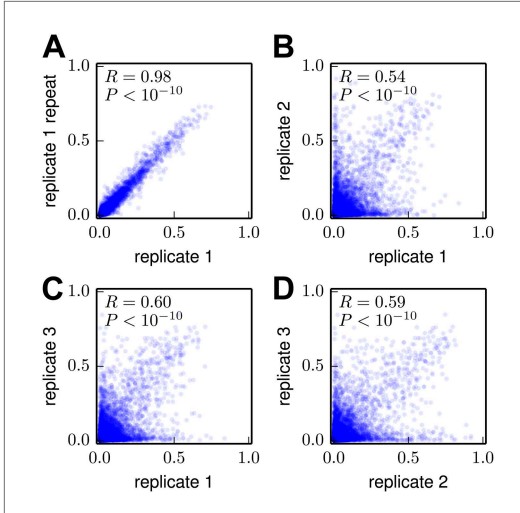

**Figure 6**. Correlations among the amino-acid preferences inferred using data from the individual biological replicates. (**A**) The preferences from two technical repeats of the sample preparation and deep sequencing of biological replicate #1 are highly correlated. (**B**)–(**D**) The preferences from the three biological replicates are substantially but imperfectly correlated. Overall, these results indicate that technical variation in sample preparation and sequencing is minimal, but that there is substantial variation between biological replicates due to stochastic differences in which mutant viruses predominate during the initial reverse-genetics step. The Pearson correlation coefficient (*R*) and associated p-value are shown in the upper-left corner of each plot. The data and code used to create this figure are available via http://jbloom.github.io/mapmuts/example_WSN_HA_2014Analysis.html; these plots are the files *correlations/replicate_1_vs_replicate_1_repeat.pdf*, *correlations/replicate_1_vs_replicate_2.pdf*, *correlations/replicate_1_vs_replicate_3.pdf*, and *correlations/replicate_2_vs_replicate_3.pdf* described therein.

## Experimental inferences are consistent with HA's natural evolution

Do the results of our deep mutational scanning experiment accurately reflect the real constraints on HA? *Table 1* uses an anecdotal comparison to a small number of existing experimental studies to suggest that they do. However, a more systematic way to address this question is to compare the inferred amino-acid preferences to the actual patterns of HA evolution in nature.

To make such a comparison, we created an alignment of HA sequences from human and swine influenza viruses descended from a common ancestor closely related to the virus that caused the 1918 influenza pandemic. *Figure 8* shows a phylogenetic tree of these sequences. The WSN HA used in our deep mutational scanning falls relatively close to the root of this tree.

The crudest comparison is simply to correlate amino-acid frequencies in the natural sequences to the experimentally inferred amino-acid preferences. *Figure 9* shows that the inferred preferences are substantially although imperfectly correlated with the natural amino-acid frequencies. However, this comparison is problematic because it fails to account for the contingent and limited sampling of mutations by natural evolution. While the deep mutational scanning is designed to sample all possible mutations, only a fraction of theoretically tolerable mutations have fixed in natural H1 HAs due to the finite timespan during which evolution has been exploring possible sequences (in other words, evolution is not at equilibrium; see *Povolotskaya and Kondrashov, 2010*). Therefore, an amino-acid frequency of close to one among the natural HA sequences in *Figure 8* might imply an absolute functional requirement for that amino acid—or it might simply mean that natural evolution has not yet happened to fix a mutation to another tolerable amino acid at that site.

A better approach is therefore to treat natural evolution as a non-equilibrium dynamic process, and ask whether the inferred amino-acid preferences accurately describe this process. This type of analysis can be done using the likelihood-based statistical framework for phylogenetics developed by Felsenstein (*1973*, *1981*). Specifically, we fix the phylogenetic tree topology to that shown in *Figure 8* and then assess the likelihood of the natural sequences given a specific evolutionary model after optimizing the branch lengths of the tree. Evolutionary models that more accurately describe HA sequence evolution will have higher likelihoods, and the relative accuracy of models can be quantified by comparing their likelihoods after correcting for the number of free parameters using AIC (*Posada and Buckley, 2004*). Previous work has described how experimental measurements of amino-acid preferences can be combined with known mutation rates to create a parameter-free phylogenetic evolutionary model from deep mutational scanning data (*Bloom, 2014*).

*Table 2* and *Table 3* compare the fit of evolutionary models based on the experimentally inferred amino-acid preferences with several existing state-of-the-art models that do not utilize this experimental information (*Goldman and Yang, 1994*; *Kosiol et al., 2007*). The model based on amino-acid

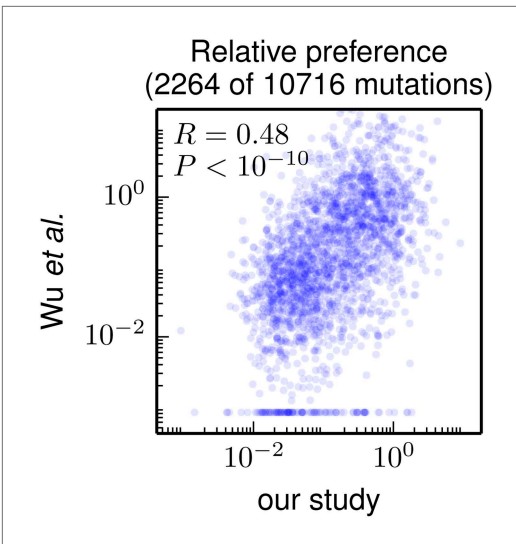

**Figure 7**. Correlation of the site-specific amino-acid preferences determined in our study with the "relative fitness" (RF) values reported by **Wu et al. (2014)**. **Wu et al. (2014)** report RF values for 2350 of the 564×19 = 10716 possible amino-acid mutations to the WSN HA examined in our study (they only examine single-nucleotide changes and disregard certain types of mutations due to oxidative damage of their DNA). To compare across the data sets, we have normalized their RF values by the RF value for the wildtype amino-acid (which they provide for only 2264 of the 2350 mutations). We then correlate on a logarithmic scale these normalized RF values with the ratio of our measurement of the preference for the mutant amino acid divided by the preference for the wildtype amino acid, using the preferences from our combined replicates. For mutations for which **Wu et al. (2014)** report an RF of zero, we assign a normalized RF equal to the smallest value for their entire data set. There is a significant Pearson correlation of 0.48 between the data sets, indicating that both our experiments and those of **Wu et al. (2014)** are capturing many of the same constraints on HA. The data and code used to create this figure are available via http://jbloom.github.io/mapmuts/example_WSN_HA_2014Analysis.html; this plot is the file *correlation_with_Wu_et_al.pdf* described therein.

preferences inferred from the combined experimental data from the three replicates describes the evolution of the naturally occurring HA sequences far better than the alternative models, despite the fact that the latter have a variety of free parameters that are optimized to improve the fit. Models based on amino-acid preferences inferred from the individual experimental replicates also fit the data better than existing models—however, the fit is poorer than for the model that utilizes the data from all three replicates. This result is consistent with the fact that the individual replicates are incomplete in their sampling of the mutational effects, meaning that aggregating the data from several replicates improves the accuracy of inferred preferences. Overall, these comparisons show that the deep mutational scanning reflects the actual constraints on HA evolution substantially better than existing quantitative evolutionary models.

## The inherent evolvability of antigenic sites on HA

The amino-acid preferences inferred from the deep mutational scanning reflect the inherent mutational tolerance of sites in HA. In contrast, the evolution of HA in nature is shaped by a combination of HA's inherent mutational tolerance and external selection pressures. Specifically, the evolution of HA in humans is strongly driven by selection for mutations that alter antigenicity (**Yewdell et al., 1979**; **Wiley et al., 1981**; **Caton et al., 1982**; **Smith et al., 2004**; **Das et al., 2013**; **Koel et al., 2013**; **Bedford et al., 2014**). The fact that such antigenic mutations fix at high frequency implies some degree of mutational tolerance at antigenic sites, since no mutations would fix if these sites were under absolute structural or functional constraint. However, it is not possible to tell from natural sequences alone whether antigenic sites are unusually mutationally tolerant compared to the rest of HA, or whether their rapid evolution is solely because they are under strong external immune selection.

To address this issue, we used the results of the deep mutational scanning to compare the inherent mutational tolerance of antigenic sites to the rest of the HA protein. **Caton et al. (1982)** mapped the antigenic sites of the H1 HA from A/Puerto Rico/8/1934 (PR8), which is closely related to the WSN HA used in our experiments. We therefore defined the 'Caton et al. antigenic sites' as the WSN residues homologous to those mapped by **Caton et al. (1982)** with the exclusion of a single site that has gained glycosylation in the WSN HA relative to the PR8 HA (see 'Materials and methods' for details). One possible concern is that **Caton et al. (1982)** mapped antigenic sites largely by selecting monoclonal-antibody escape mutants, and so these sites might be biased towards being more mutationally tolerant. We therefore also made a broader classification of 'antigenic sites and contacting residues' consisting of the Caton et al. antigenic sites *plus* all surface-exposed residues in contact with these sites (see 'Materials and methods' for details). This broader classification includes all residues in regions of the HA surface targeted by antibodies, and so

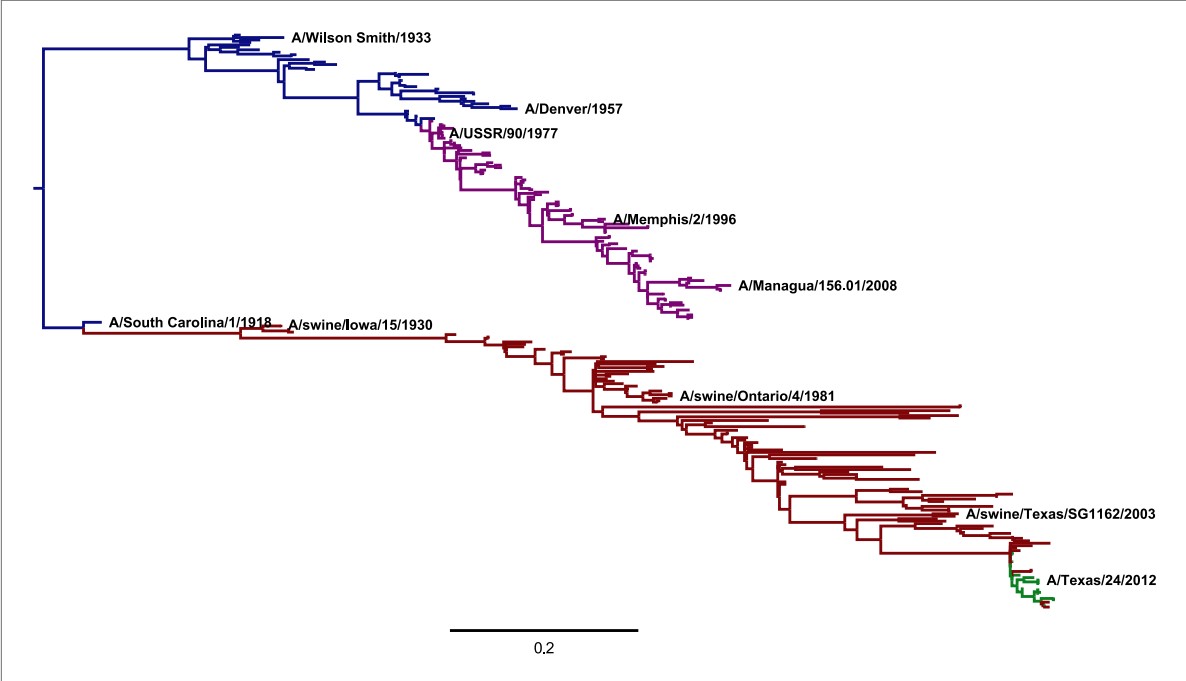

**Figure 8**. A phylogenetic tree of human and swine H1 HA sequences descended from a common ancestor closely related to the 1918 virus. The WSN virus used in the experiments here is a lab-adapted version of the *A/Wilson Smith/1933* strain. Human H1N1 that circulated from 1918 until 1957 is shown in blue. Human seasonal H1N1 that reappeared in 1977 is shown in purple. Swine H1N1 is shown in red. The 2009 pandemic H1N1 is shown in green. This tree was constructed using *codonPhyML* (***Gil et al., 2013***) with the substitution model of ***Goldman and Yang (1994)***. This plot is the file *CodonPhyML_Tree_H1_HumanSwine_GY94/annotated_tree.pdf* described at http://jbloom.github.io/phyloExpCM/example_ 2014Analysis_Influenza_H1_HA.html. ***Figure 8—figure supplement 1*** shows a tree estimated for the same sequences using the substitution model of ***Kosiol et al. (2007)***.

The following figure supplements are available for figure 8:

**Figure supplement 1**. A phylogenetic tree of the same sequences shown in ***Figure 8***, this time inferred using the substitution model of ***Kosiol et al. (2007)***.

should not be biased by whether sites are amenable to the selection of monoclonal-antibody escape mutants. We hypothesized that both sets of antigenic sites would have unusually high mutational tolerance.

For comparison, we used two classifications of receptor-binding residues ('Materials and methods'). The first classification consists of residues that have important roles in receptor binding (***Martin et al., 1998***) *and* are conserved in H1 HAs; these residues are mostly deep in the binding pocket. The second classification consists of all residues that contact the sialic-acid receptor in the crystal structure, regardless of their level of conservation. We hypothesized that the core set of conserved receptor-binding residues would have unusually low mutational tolerance, but that the set of all receptor-binding residues would have typical levels of mutational tolerance since influenza routinely escapes from antibodies that target the periphery of the receptor-binding pocket (***Koel et al., 2013***).

The positions of the Caton et al. antigenic sites and the conserved receptor-binding residues in the primary sequence are indicated by the top overlay bar in ***Figure 5***. Visual inspection suggests that the conserved receptor-binding residues are indeed relatively intolerant of mutations (have a strong preference for one specific amino acid), whereas the Caton et al. antigenic sites are relatively tolerant of mutations (have roughly equivalent preferences for many amino acids).

For a more quantitative analysis, we computed a site entropy from the inferred amino-acid preferences—larger site entropies indicate a higher inherent tolerance for mutations. The site entropies of all residues are displayed on the HA protein structure in ***Figure 10***. Visual inspection suggests that both classifications of antigenic sites have unusually high mutational tolerance, whereas the conserved receptor-binding residues have unusually low mutational tolerance.

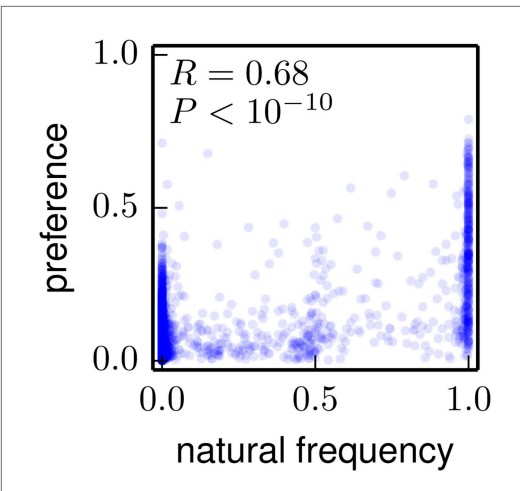

**Figure 9**. The frequencies of amino acids among the naturally occurring HA sequences in *Figure 8* vs the amino-acid preferences inferred from the combined replicates (*Figure 5*). Note that a natural frequency close to one or zero could indicate absolute selection for or against a specific amino acid, but could also simply result from the fact that natural evolution has not completely sampled all possible mutations compatible with HA structure and function. The Pearson correlation coefficient ($R$) and associated p-value are shown on the plot. This plot is the file *natural_frequency_vs_preference.pdf* described at http://jbloom.github.io/phyloExpCM/example_2014Analysis_Influenza_H1_HA.html.

We next tested whether these visual observations were supported by a rigorous statistical analysis. A confounding factor in comparing mutational tolerance across different sets of residues is that sites with higher solvent accessibility are typically more tolerant of mutations (*Bustamante et al., 2000*; *Ramsey et al., 2011*). To correct for this fact, we computed the relative solvent accessibility (RSA) for all residues in the HA crystal structure. Residues with RSAs close to zero are buried and are expected to be fairly intolerant of mutations, whereas residues with RSAs substantially greater than zero are surface exposed and are expected to be fairly tolerant of mutations. *Figure 10* plots site entropy as a function of RSA for HA1 residues. This figure shows that sites with higher RSA are more mutationally tolerant as expected. However, the figure also suggests that both classifications of antigenic sites are more mutationally tolerant than other residues with equivalent RSA. The figure also suggest that the conserved receptor-binding residues are less mutationally tolerant than other residues with equivalent RSA, whereas the set of all receptor-binding residues have fairly typical mutational tolerance. These observations are supported by the statistical analyses in *Table 4*: even after correcting for RSA, there is a significant trend for antigenic sites to have high mutational tolerance, and for conserved receptor-binding residues to have low mutational tolerance.

Overall, these results show that antigenic sites in HA have unusually high inherent mutational tolerance, suggesting that this property combines with external immune selection to contribute to HA's rapid antigenic evolution. These results also show that while a core group of conserved residues deep in the receptor-binding pocket have unusually low mutational tolerance, the bulk of residues that contact the receptor are not under exceptional constraint. This fact probably explains why HA is able to escape from antibodies targeting the periphery of the receptor-binding pocket (*Koel et al., 2013*), and why only rare antibodies that penetrate deep into this pocket are broadly neutralizing (*Whittle et al., 2011*).

## HA's antigenic evolvability is not shared by all influenza proteins

The foregoing results show that the antigenic sites in HA have an unusually high inherent tolerance for mutations. Is this antigenic evolvability an exceptional feature of HA, or is it commonly shared by other viral proteins? Ideally one would compare HA to the major surface antigens of other viruses with high (e.g., HIV) and low (e.g., measles) rates of antigenic evolution—but unfortunately comparable data sets for these other viruses are not yet available. Therefore, we instead compared the antigenic evolvability of HA to that of influenza nucleoprotein (NP), a protein for which we have recently performed a similar deep mutational scanning experiment (*Bloom, 2014*).

The adaptive immune system targets NP via cytotoxic T-lymphocytes (CTLs) (*Valkenburg et al., 2011*). Although the selection exerted by these CTLs is believed to be weaker than the antibody-mediated selection on HA's antigenic sites (*Bhatt et al., 2011*), influenza does benefit from mutations in NP that promote escape from CTLs (*Berkhoff et al., 2007*; *Valkenburg et al., 2013*). However, whereas HA rapidly evolves to escape from antibodies, NP does not appear to have any special propensity for rapid evolution of the epitopes targeted by CTLs. Instead, mutations in NP's CTL epitopes are often deleterious and require secondary permissive or compensatory mutations to fix without a fitness cost (*Rimmelzwaan et al., 2004*; *Berkhoff et al., 2005*, *2006*; *Gong et al., 2013*).

**Table 2.** An evolutionary model derived from the experimentally inferred amino-acid preferences describes the HA sequence phylogeny in *Figure 8* far better than a variety of existing state-of-the-art models

| Model | Δ AIC | Log likelihood | Parameters (optimized + empirical) |
|---|---|---|---|
| Combined | 0.0 | −24088.7 | 0 (0 + 0) |
| replicate 3 | 303.2 | −24240.3 | 0 (0 + 0) |
| Combined, Halpern and Bruno | 500.6 | −24339.0 | 0 (0 + 0) |
| replicate 1 | 535.4 | −24356.4 | 0 (0 + 0) |
| replicate 3, Halpern and Bruno | 657.8 | −24417.6 | 0 (0 + 0) |
| replicate 2 | 876.2 | −24526.8 | 0 (0 + 0) |
| GY94, gamma $\omega$, gamma rates | 882.6 | −24517.0 | 13 (4 + 9) |
| replicate 1, Halpern and Bruno | 983.2 | −24580.3 | 0 (0 + 0) |
| GY94, gamma $\omega$, one rate | 1109.7 | −24631.5 | 12 (3 + 9) |
| replicate 2, Halpern and Bruno | 1190.0 | −24683.7 | 0 (0 + 0) |
| KOSI07, gamma $\omega$, gamma rates | 1620.5 | −24834.9 | 64 (4 + 60) |
| GY94, one $\omega$, gamma rates | 1859.4 | −25006.4 | 12 (3 + 9) |
| KOSI07, gamma $\omega$, one rate | 1883.0 | −24967.2 | 63 (3 + 60) |
| KOSI07, one $\omega$, gamma rates | 2378.8 | −25215.1 | 63 (3 + 60) |
| GY94, one $\omega$, one rate | 2544.5 | −25350.0 | 11 (2 + 9) |
| KOSI07, one $\omega$, one rate | 3040.0 | −25546.7 | 62 (2 + 60) |
| combined, randomized | 5632.8 | −26905.1 | 0 (0 + 0) |
| replicate 1, randomized | 6002.4 | −27089.9 | 0 (0 + 0) |
| replicate 3, randomized | 6138.8 | −27158.1 | 0 (0 + 0) |
| replicate 2, randomized | 6477.8 | −27327.6 | 0 (0 + 0) |
| combined, randomized, Halpern and Bruno | 7072.8 | −27625.1 | 0 (0 + 0) |
| replicate 1, randomized, Halpern and Bruno | 7795.0 | −27986.2 | 0 (0 + 0) |
| replicate 3, randomized, Halpern and Bruno | 7891.8 | −28034.6 | 0 (0 + 0) |
| replicate 2, randomized, Halpern and Bruno | 8494.4 | −28335.9 | 0 (0 + 0) |

The model is most accurate if it utilizes data from the combined experimental replicates, but it also outperforms existing models even if the data are only derived from individual replicates. Models are ranked by AIC (***Posada and Buckley, 2004***). *GY94* indicates the model of ***Goldman and Yang (1994)***, and *KOSI07* indicates the model of ***Kosiol et al. (2007)***. The nonsynonymous/synonymous ratio ($\omega$) and the substitution rate are either estimated as a single value or drawn from a four-category gamma distribution. Randomizing the experimentally inferred preferences among sites makes the models far worse. The models work best fixation probabilities are computed from the preferences using the first equation proposed in ***Bloom (2014)***. The table also shows the results if the fixation probabilities are instead computed using the equation of ***Halpern and Bruno (1998)*** as described in ***Bloom (2014)***. This table is the file *H1_HumanSwine_GY94_summary.tex* described at http://jbloom.github.io/phyloExpCM/example_2014Analysis_Influenza_H1_HA.html. ***Table 3*** shows the results when the tree topology is instead estimated using the substitution model of ***Kosiol et al. (2007)***.

Therefore, we hypothesized that unlike HA's highly evolvable antigenic sites, NP's CTL-antigenic sites would *not* possess unusually high inherent mutational tolerance.

To test this hypothesis, we used a previously described delineation of epitopes in NP from the human H3N2 strain A/Aichi/2/1968 with experimentally validated human CTL responses (***Gong and Bloom, 2014***). In this delineation, less than a quarter of NP's sites participate in multiple CTL epitopes. We used the results of our previous deep mutational scanning of NP to compare the inherent mutational tolerance of sites that participate in multiple CTL epitopes to all other sites in NP. As shown in *Figure 11* and *Table 5*, the NP sites involved in multiple CTL epitopes have an inherent mutational tolerance that is indistinguishable from other sites in the protein. Therefore, NP does not possess any

**Table 3.** An evolutionary model derived from the experimentally inferred amino-acid preferences also outperforms existing models for the tree topology in *Figure 8—figure supplement 1*

| Model | ΔAIC | Log likelihood | Parameters (optimized + empirical) |
|---|---|---|---|
| Combined | 0.0 | −24082.5 | 0 (0 + 0) |
| replicate 3 | 304.8 | −24234.9 | 0 (0 + 0) |
| Combined, Halpern and Bruno | 494.4 | −24329.7 | 0 (0 + 0) |
| replicate 1 | 534.2 | −24349.6 | 0 (0 + 0) |
| replicate 3, Halpern and Bruno | 653.2 | −24409.1 | 0 (0 + 0) |
| replicate 2 | 869.4 | −24517.2 | 0 (0 + 0) |
| GY94, gamma $\omega$, gamma rates | 876.7 | −24507.8 | 13 (4 + 9) |
| replicate 1, Halpern and Bruno | 976.8 | −24570.9 | 0 (0 + 0) |
| GY94, gamma $\omega$, one rate | 1101.0 | −24621.0 | 12 (3 + 9) |
| replicate 2, Halpern and Bruno | 1180.4 | −24672.7 | 0 (0 + 0) |
| KOSI07, gamma $\omega$, gamma rates | 1609.0 | −24823.0 | 64 (4 + 60) |
| GY94, one $\omega$, gamma rates | 1856.2 | −24998.6 | 12 (3 + 9) |
| KOSI07, gamma $\omega$, one rate | 1867.3 | −24953.1 | 63 (3 + 60) |
| KOSI07, one $\omega$, gamma rates | 2367.9 | −25203.4 | 63 (3 + 60) |
| GY94, one $\omega$, one rate | 2548.3 | −25345.6 | 11 (2 + 9) |
| KOSI07, one $\omega$, one rate | 3028.0 | −25534.5 | 62 (2 + 60) |
| Combined, randomized | 5628.0 | −26896.5 | 0 (0 + 0) |
| replicate 1, randomized | 5993.6 | −27079.3 | 0 (0 + 0) |
| replicate 3, randomized | 6138.0 | −27151.5 | 0 (0 + 0) |
| replicate 2, randomized | 6475.2 | −27320.1 | 0 (0 + 0) |
| combined, randomized, Halpern and Bruno | 7069.4 | −27617.2 | 0 (0 + 0) |
| replicate 1, randomized, Halpern and Bruno | 7786.8 | −27975.9 | 0 (0 + 0) |
| replicate 3, randomized, Halpern and Bruno | 7889.2 | −28027.1 | 0 (0 + 0) |
| replicate 2, randomized, Halpern and Bruno | 8496.0 | −28330.5 | 0 (0 + 0) |

This table differs from *Table 2* in that it uses the tree topology inferred with the model of *Kosiol et al. (2007)* rather than *Goldman and Yang (1994)*. This table is the file *H1_HumanSwine_KOSI07_summary.tex* described at http://jbloom.github.io/phyloExpCM/example_2014Analysis_Influenza_H1_HA.html.

special inherent mutational tolerance in its CTL epitopes. This finding implies that a high level of antigenic evolvability is not a general feature of all viral proteins, but is instead at least somewhat unique to HA.

## Discussion

A fundamental challenge in studying the natural evolution of influenza is separating the effects of external selection pressures from inherent structural and functional constraints. The evolutionary patterns observed in natural sequences are shaped by a combination of inherent mutational tolerance and external pressures such as immune selection, and the analysis of such sequences is further confounded by the fact that influenza is not at evolutionary equilibrium.

Here we have quantified the inherent mutational tolerance of influenza HA by using deep mutational scanning (*Fowler et al., 2010*; *Araya and Fowler, 2011*) to simultaneously assess the impact on viral growth of the vast majority of the ≈$10^4$ possible amino-acid mutations to influenza HA. The information obtained from the deep mutational scanning is consistent with existing knowledge about the effects of mutations on HA function and structure. For instance, the deep mutational scanning shows strong selection for specific amino acids known to play important roles in HA's receptor-binding activity, fusion activity, and proteolytic activation (*Martin et al., 1998*; *Qiao et al., 1999*; *Stech et al., 2005*). Similarly, at the sites of known temperature-sensitive mutations to HA (*Nakajima et al., 1986*), the deep mutational scanning identifies the more stabilizing amino-acid as more favorable.

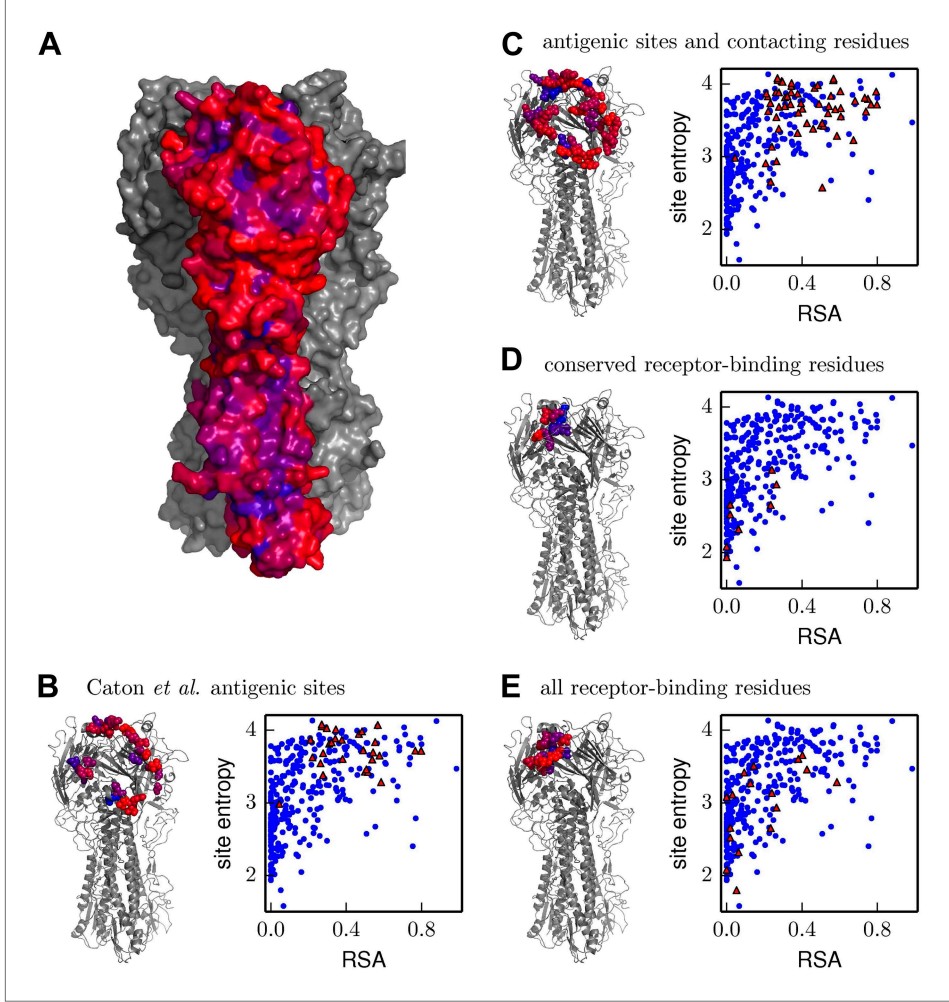

**Figure 10**. Inherent mutational tolerance of HA's receptor-binding residues and antigenic sites. (**A**) Surface of HA with one monomer colored by site entropy as determined by the deep mutational scanning; blue indicates low mutational tolerance and red indicates high mutational tolerance. (**B**) The structure shows residues classified as antigenic sites by *Caton et al. (1982)* in colored spheres; the plot shows site entropy vs relative solvent accessibility (RSA) of these residues (red triangles) and all other HA1 residues in the crystal structure (blue circles). (**C**) Antigenic sites of *Caton et al. (1982)* plus all other surface-exposed residues that contact these sites. (**D**) Conserved receptor-binding residues. (**E**) All receptor-binding residues. *Table 4* shows that residues in (**B**) and (**C**) have unusually high mutational tolerance, residues in (**D**) have unusually low mutational tolerance, and residues in (**E**) do not have unusual mutational tolerance. The data and code to create all panels of this figure is provided via http://jbloom.github.io/mapmuts/example_WSN_HA_2014Analysis.html. The structure is PDB 1RVX (*Gamblin et al., 2004*).

Broader trends from the deep mutational scanning are also in agreement with current thinking about mutational effects. For example, the deep mutational scanning finds that there is strong purifying selection against stop-codon mutations and many nonsynonymous mutations, but that there is only weak selection against synonymous mutations. All of these results suggest that the deep mutational scanning faithfully captures both the specific and general effects of mutations on HA.

The comprehensive information generated by the deep mutational scanning can be used to create quantitative evolutionary models for analyzing HA sequence phylogenies. Here we have shown that an evolutionary model constructed from our deep mutational scanning data describes the evolution of human and swine H1 HAs far better than existing state-of-the-art models for sequence evolution. We anticipate that separating HA's inherent mutational tolerance from external selection should also eventually allow the external selection pressures to be studied in greater detail. For example, one might imagine that sites in HA that exhibit evolutionary patterns that deviate from the quantitative

**Table 4.** The antigenic sites are more significantly mutationally tolerant than other HA1 residues with similar relative solvent accessibility (RSA), the conserved receptor-binding residues are significantly less mutationally tolerant than other similar residues, and sites in the more expansive set of all receptor-binding residues have typical levels of mutational tolerance

Model: site entropy ~ RSA + (Caton et al. antigenic site) + intercept

| Property | Estimate | Standard error | p-value |
|---|---|---|---|
| RSA | 1.29 | 0.12 | $<10^{-10}$ |
| Caton et al. antigenic site | 0.30 | 0.09 | $1.6 \times 10^{-3}$ |

Model: site entropy ~ RSA + (antigenic site or contacting residue) + intercept

| Property | Estimate | Standard error | p-value |
|---|---|---|---|
| RSA | 1.22 | 0.13 | $<10^{-10}$ |
| antigenic site or contacting residue | 0.23 | 0.07 | $2.2 \times 10^{-3}$ |

Model:   site entropy ~ RSA + (conserved receptor binding) + intercept

| Property | Estimate | Standard error | p-value |
|---|---|---|---|
| RSA | 1.38 | 0.11 | $<10^{-10}$ |
| conserved receptor binding | −0.52 | 0.16 | $1.7 \times 10^{-3}$ |

Model:   site entropy ~ RSA + (all receptor binding) + intercept

| Property | Estimate | Standard error | p-value |
|---|---|---|---|
| RSA | 1.40 | 0.11 | $<10^{-10}$ |
| all receptor binding | −0.18 | 0.11 | 0.12 |

The sets of residues analyzed here are those shown in **Figure 10**. Shown here are the results of multiple linear regression of the continuous dependent variable of site entropy (as computed from the amino-acid preferences) vs the continuous independent variable of RSA and the binary variable of being a receptor-binding residue or being an antigenic site. The data and code used to perform these analyses are available via http://jbloom.github.io/mapmuts/example_WSN_HA_2014Analysis.html.

model created from our deep mutational scanning are likely to be under external selection. Future work that augments deep mutational scanning with specific experimentally defined selection pressures (such as antibodies against HA) could aid in further elucidation of the forces that shape influenza evolution. It also may be possible to utilize high-throughput experimental data on mutational effects to better estimate the fitness of naturally occurring strains in a way that aids in prediction of the year-to-year strain dynamics of influenza (*Łuksza and Lässig, 2014*).

The deep mutational scanning also enabled us to assess the extent to which HA's inherent mutational tolerance contributes to influenza's antigenic evolvability. It remains a mystery why error-prone RNA viruses differ so widely in their capacity for evolutionary escape from immunity, with some (e.g., influenza and HIV) undergoing rapid antigenic evolution while others (e.g., measles) show little antigenic change on relevant timescales (*Lipsitch and O'Hagan, 2007*; *Koelle et al., 2006*; *Heaton et al., 2013*). Our data demonstrate that the antigenic sites in HA are unusually tolerant to mutations, implying that inherent evolutionary plasticity at sites targeted by the immune system is one factor that contributes to influenza's rapid antigenic evolution. This high mutational tolerance at antigenic sites could itself be a property that influenza has evolved to aid in its antigenic escape—or it might simply be an unfortunate coincidence that the immune system focuses on especially plastic portions of HA. In either case, it is intriguing to speculate whether a high inherent mutational tolerance in antigenic sites is also a feature of other antigenically variable RNA viruses. Application of the deep mutational scanning approach used here to additional viruses should provide a means to address this question.

## Materials and methods

### Availability of data and computer code

Illumina sequencing data are available at the SRA, accession SRP040983 (http://www.ncbi.nlm.nih.gov/sra/?term=SRP040983). Source code and a description of the computational process used to

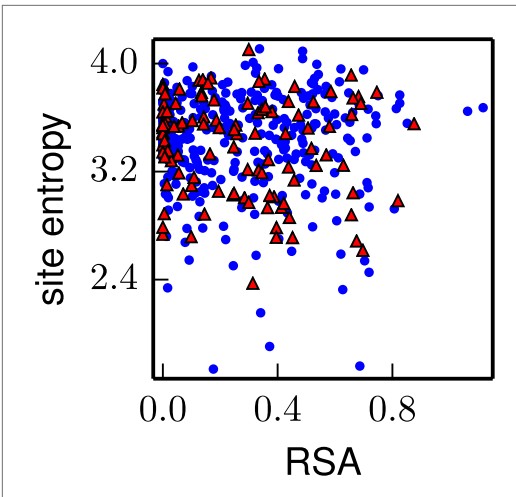

**Figure 11**. The inherent mutational tolerance of NP's CTL epitopes is indistinguishable from that of non-epitope sites in NP. The plot shows the site entropy vs relative solvent accessibility (RSA) of NP residues that participate in multiple CTL epitopes (red triangles) and all other NP residues in the crystal structure (blue circles). Visual inspection suggests that the epitope sites have mutational tolerance comparable to other sites, and this result is supported by the statistical analysis in *Table 5*. Note that unlike for HA, there is no trend for RSA to correlate with site entropy—this could be because many of NP's surface-exposed sites are constrained by interactions with viral RNA. The CTL epitopes are those delineated in the first supplementary table of *Gong and Bloom (2014)*. The site entropies are computed from a previously described deep mutational scan of NP, and are the values in the first supplementary file of *Bloom (2014)*; the RSA values are also taken from that reference. The data and code used to generate this plot is available via http://jbloom.github.io/mapmuts/example_WSN_HA_2014Analysis.html; the plot itself is the file *NP_CTL_entropy_rsa_correlation.pdf* described therein.

analyze the sequencing data and infer the amino-acid preferences is at http://jbloom.github.io/mapmuts/example_WSN_HA_2014Analysis.html. Source code and a description of the computational process used for the phylogenetic analyses is available at http://jbloom.github.io/phyloExpCM/example_2014Analysis_Influenza_H1_HA.html.

## HA sequence numbering

A variety of different numbering schemes for HA are used in the literature. Unless noted otherwise, residues are numbered here using sequential numbering of the WSN HA protein sequence (*Supplementary file 1*) starting with one at the N-terminal methionine. In some cases, the number of the corresponding residues in the widely used H3 numbering scheme is also indicated. These numbering systems can be interconverted using the Python script available at https://github.com/jbloom/HA_numbering.

## Generation of HA codon mutation library

The HA codon-mutant library was generated using the oligo-based PCR mutagenesis protocol described previously by *Bloom (2014)*. The only differences from that protocol were that HA was used as the template rather than NP, and that only two overall rounds of mutagenesis were performed, rather than the three rounds used by *Bloom (2014)*. This reduction in the number of rounds of mutagenesis reduced the average number of codon mutations from the ≈ three per clone in *Bloom (2014)* to the ≈ two per clone shown in *Figure 2*. The libraries were created in full biological triplicate, meaning that each experimental replicate was derived from an independent plasmid mutant library.

The end primers for the mutagenesis were 5'-cgatcacgtctctgggagcaaaagcaggggaaaataaaaacaac-3' and 5'-gatacacgtctcatattagtagaaacaagggtgtttttccttatatttctg-3' (these primers include BsmBI restriction sites). The mutagenic primers were ordered from Integrated DNA Technologies, and are listed in *Supplementary file 2*.

The final products from the codon mutagenesis PCR were gel purified and digested with BsmBI (R0580L; New England Biolabs, Ipswich, Massachusetts). The BsmBI-digested HA was ligated into a dephosphorylated (Antarctic Phosphatase, M0289L; New England Biolabs) and BsmBI-digested preparation of the bidirectional reverse-genetics plasmid pHW2000 (*Hoffmann et al., 2000*) using T4 DNA ligase (M0202S; New England Biolabs). Column-purified ligations were electroporated into ElectroMAX DH10B T1 phage-resistant competent cells (12033-015; Invitrogen, Carlsbad, California) and plated on LB plates containing 100 μg/ml of ampicilin. A 1:4000 dilution of each transformation was plated in parallel to enable estimation of the number of unique transformants—we obtained at least two-million unique colonies per transformation. For each replicate of the codon-mutant library, we performed three transformations to generate approximately six-million independent clones per replicate library. Control ligations lacking an insert yielded at least 100 times fewer colonies, indicating a very low rate of background self-ligation of the pHW2000 plasmid. The transformants from each HA mutant library replicate were pooled, cultured in LB supplemented with ampicillin, and mini-prepped to generate the HA codon mutant plasmid libraries.

**Table 5.** There is no statistically significant difference between the inherent mutational tolerance of NP sites involved in multiple CTL epitopes and all other NP residues

**Model: NP site entropy ~ RSA + (multiple CTL epitopes) + intercept**

| Property | Estimate | Standard error | p-value |
|---|---|---|---|
| RSA | −0.05 | 0.07 | 0.52 |
| multiple CTL epitopes | −0.04 | 0.04 | 0.31 |

The table shows the result of multiple linear regression of the continuous dependent variable of site entropy (as computed from the amino-acid preferences) vs the continuous independent variable of RSA and the binary variable of participating in multiple CTL epitopes. The data set analyzed here is plotted in **Figure 11**. The data and code used to perform this analysis are available via http://jbloom.github.io/mapmuts/example_WSN_HA_2014Analysis.html.

For the Sanger sequencing analysis shown in **Figure 2**, we picked and prepped 34 independent colonies for sequencing. The full analysis of this Sanger sequencing is available at https://github.com/jbloom/SangerMutantLibraryAnalysis/tree/v0.2.

## Virus rescue and passage in cells

The HA mutant plasmid libraries were used to generate pools of mutant influenza viruses by reverse genetics (**Hoffmann et al., 2000**). Cocultures of 293T and MDCK-SIAT1 cells were transfected with equal amounts of HA (either unmutated or one of the plasmid mutant libraries) cloned into pHW2000 as described above, plus the seven other WSN genes in bidirectional reverse-genetics plasmids (pHW181-PB2, pHW182-PB1, pHW183-PA, pHW185-NP, pHW186-NA, pHW187-NA, pHW188-NS), which were kind gifts from Robert Webster of St. Jude Children's Research Hospital. Overall, six viral rescues and passages were performed, each using a different HA plasmid preparation: the three HA mutant library replicates (eventually yielding the **mutvirus** samples in **Figure 1**) and three independent unmutated HAs (eventually yielding the **virus** samples in **Figure 1**).

Each of the viral rescues was performed by transfecting multiple wells of cells in an effort to increase the diversity of the rescued viruses. Specifically, two 12-well dishes were transfected per rescue. Cells were plated at $2 \times 10^5$ 293T cells and $5 \times 10^4$ MDCK-SIAT1 cells per well in D10 (DMEM supplemented with 10% heat-inactivated FBS, 2 mM L-glutamine, 100 U of penicillin/ml, and 100 µg of streptomycin/ml), and then each well was transfected with 1 µg of total plasmid DNA (125 ng of each of the eight plasmids) using the BioT transfection reagent (Bioland B01-02, Paramount, California). At 12 to 18 hr post-transfection, the medium was changed to our WSN viral growth media: Opti-MEM supplemented with 0.5% heat-inactivated FBS, 0.3% BSA, 100 U of penicillin/ml, 100 µg of streptomycin/ml, and 100 µg of calcium chloride/ml. This media does not contain trypsin since viruses with the WSN HA and NA are trypsin independent (**Goto and Kawaoka, 1998**). Viral supernatants were collected 72 hr post-transfection, and the supernatants from the different wells were pooled for each viral rescue. These pooled supernatants were then clarified by centrifugation at 2000×*g* for 5 min, aliquoted, and frozen at −80 ℃. Aliquots were then thawed and titered by TCID50 (see below).

For viral passage, each viral rescue replicate was passaged in four 10-cm dishes. Briefly, $6 \times 10^6$ MDCK-SIAT1 cells per 10-cm dish in *WSN viral growth media* were infected with $6\times10^5$ infectious particles (multiplicity of infection of 0.1). Since there are four dishes for each replicate, this maintains a diversity of $2.4 \times 10^6$ TCID50 units per replicate. The passaged viral supernatants were collected at 50 hr post-infection, and the supernatants for the four plates were pooled for each replicate. These pooled supernatants were clarified at 2000 × g for 5 min, aliquoted, and frozen at −80 ℃. Aliquots were then thawed and titered by TCID50.

## Virus titering by TCID50

The viruses were titered by TCID50 (50% tissue culture infectious dose). In this assay, 10 µl of a 1:10 dilution of the viral supernatant to be titered was added to the first row of a 96-well tissue culture plate containing 90 µl of WSN viral growth media. At least one no-virus control supernatant was included on each plate as a negative control. The virus was then serially diluted 1:10 down the rows of the plates, and then $5 \times 10^3$ MDCK-SIAT1 cells were added to each well. The plates were then incubated at 37℃, and scored for cytopathic effects caused by viral growth after for 65–72 hr. Virus titers were calculated

by the method of *Reed and Muench (1938)* implemented via the Python script at https://github.com/jbloom/reedmuenchcalculator.

## Generation of samples for Illumina deep sequencing

The deep sequencing samples were prepared from PCR amplicons that were generated exactly as described for the **DNA**, **mutDNA**, **virus**, and **mutvirus** samples in *Bloom (2014)*. The viral RNA template for the **virus** and **mutvirus** were isolated using freshly purchased Trizol reagent (15596-026; Life Technologies) in order to avoid any oxidative damage associated with old reagents. After performing reverse transcription as described in *Bloom (2014)*, quantitative PCR (qPCR) was used to quantify the number of HA cDNA molecules to ensure that there were at least $10^6$ unique template molecules before beginning the subsequent PCR amplification. The qPCR primers were designed based on those described by *Marsh et al. (2007)*, and were 5'-taacctgctcgaagacagcc-3' and 5'-agagccatccggtgatgtta-3'.

The PCR amplicons were fragmented and barcoded using the custom modification of Illumina's Nextera kit using the protocol described in *Bloom (2014)*. Samples were barcoded as follows: **DNA**–N701, **mutDNA**–N702, **virus**–N704, and **mutvirus**–N705. For each of the three biological replicates, these four samples were pooled and sequenced on their own Illumina lane with 50-nucleotide paired-end reads as described in *Bloom (2014)*. For the technical sequencing repeat of biological replicate #1, the library preparation and sequencing were repeated from the same viral RNA templates. This technical repeat therefore only quantifies variation associated with sample preparation and sequencing, whereas the biological replicates also quantify variation associated with the processes of codon-mutant library creation, virus generation, and virus passage.

## Analysis of deep sequencing data

The deep sequencing data was analyzed using the *mapmuts* computer program (*Bloom, 2014*). A description of the analysis approach and the resulting data files and figures produced are available at http://jbloom.github.io/mapmuts/example_WSN_HA_2014Analysis.html.

Briefly, paired reads were overlapped as illustrated in *Figure 3—figure supplement 1* and then aligned to HA. Reads were retained only if both reads in the pair passed the default Illumina filter, had average Q-scores of at least 25, overlapped for at least 30 nucleotides with no more than one mismatch, and the overlap aligned to the HA gene with no more than six mismatches. *Figure 3—figure supplement 2* shows the number of reads for each sample that met these criteria. Most reads that did not meet these criteria failed to do so because they could not be paired with at least 30 nucleotides of overlap—a situation that arises when the HA fragment produced by the Nextera fragmentation produces a fragment smaller than 30 nucleotides or larger than 70 nucleotides. Codon identities were called only if both overlapped paired reads agreed on the identity of the codon. This requirement reduces the error rate, because it is rare for both paired reads to independently experience the same sequencing error.

As shown in *Figure 4*, we estimated that 85% of possible codon mutations were sampled at least five times by the mutant viruses. To estimate the fraction of amino-acid mutations that would have been sampled, we simulated randomly selecting 85% of the mutant codons from the HA sequence, and determined that these codons encoded ≈97% of the amino-acid mutations.

## Inference of amino-acid preferences and site entropies

The counts of each codon identity in the deep sequencing data was used to infer the 'preference' of each site for each amino acid as described in *Bloom (2014)*. This inference was also done using the *mapmuts* computer program as detailed at http://jbloom.github.io/mapmuts/example_WSN_HA_2014Analysis.html.

Briefly, the preference $\pi_{r,a}$ of site $r$ for amino-acid $a$ represents the expected frequency of that amino acid in a hypothetical library where each amino-acid is introduced at equal frequency. Specifically, the expected frequency $f_{r,x}^{\textbf{mutvirus}}$ of mutant codon $x$ at site $r$ in the **mutvirus** sample is related to the preference for its encoded amino-acid $\mathcal{A}(x)$ by

$$f_{r,x}^{\textbf{mutvirus}} = \varepsilon_{r,x} + \rho_{r,x} + \frac{\mu_{r,x} \times \pi_{r,\mathcal{A}(x)}}{\sum\limits_{y} \mu_{r,y} \times \pi_{r,\mathcal{A}(y)}},$$

where $\varepsilon_{r,x}$ is the rate at which site $r$ is erroneously read to be codon $x$, $\rho_{r,x}$ is the rate at which site $r$ is erroneously reverse-transcribed to codon $x$, and $\mu_{r,x}$ is the rate at which site $r$ is mutagenized to

codon $x$ in the mutant DNA sample. These unknown error and mutation rate parameters are inferred from the **DNA**, **virus**, and **mutvirus** samples using the Bayesian approach described in *Bloom (2014)*. Inferences of the posterior mean preferences $\pi_{r,a}$ were made separately for each replicate of the experiment, and the correlations among these inferences from different replicates are in *Figure 6*. The final 'best' inferred preferences from the combined data of the three biological replicates were obtained by averaging the preferences obtained from the three biological replicates. These final inferred preferences are provided in *Supplementary file 3* and displayed graphically in *Figure 5*.

The site entropies in *Figure 10* and *Table 4* were calculated from the amino-acid preferences as $h_r = \sum_a \pi_{r,a} \times \log_2 \pi_{r,a}$. These site entropies are therefore in bits. Higher site entropies indicate a higher inherent mutational tolerance.

## Alignment of naturally occurring HAs and phylogenetic tree

The inferred amino-acid preferences were compared to amino-acid frequencies in an alignment of naturally occurring H1N1 HAs from swine and human lineages descended from a close relative of the 1918 virus. Briefly, all full-length H1 HAs from these hosts were downloaded from the Influenza Virus Resource (*Bao et al., 2008*). Up to three sequences per host and year were randomly subsampled and used to build a phylogenetic tree. Clear outliers from the molecular clock (typically lab artifacts or mis-annotated sequences) were iteratively excluded and the trees were rebuilt. The final sequence alignment is in *Supplementary file 4*. This alignment was used to build the phylogenetic trees in *Figure 8* and *Figure 8—figure supplement 1* with *codonPhyML* (*Gil et al., 2013*) using the codon-substitution model of (*Goldman and Yang, 1994*) or (*Kosiol et al., 2007*) with empirical codon frequencies determined using the CF3x4 method (*Pond et al., 2010*) or the *F* method, respectively. In both cases, the nonsynonymous-synonymous ratio ($\omega$) was drawn from four gamma-distributed categories (*Yang et al., 2000*). A description of this process is at http://jbloom.github.io/phyloExpCM/example_2014Analysis_Influenza_H1_HA.html.

## Comparison of evolutionary models

We compared the accuracy with which the naturally occurring HA phylogeny was described by an evolutionary model based on the experimentally measured amino-acid preferences vs several standard codon-substitution models. These comparisons were used made using *HYPHY* (*Pond et al., 2005*) and *phyloExpCM* (*Bloom, 2014*). A description of this analysis is at http://jbloom.github.io/phyloExpCM/example_2014Analysis_Influenza_H1_HA.html.

Briefly, the phylogenetic tree topology was fixed to that shown in *Figure 8* or *Figure 8—figure supplement 1*. The branch lengths and any free parameters of the evolutionary model were then optimized by maximum likelihood. The experimentally determined evolutionary models were constructed from the inferred amino-acid preferences reported here and the experimentally measured mutation rates reported in *Bloom (2014)*. The 'fixation probabilities' were computed using either the Metropolis-like relationship described in *Bloom (2014)* or the relationship proposed by *Halpern and Bruno (1998)*. The results of these comparisons are in *Tables 2 and 3*. All of these comparisons show that the experimentally determined evolutionary models are far superior to the various standard models.

## Structural analyses

The WSN HA studied here has a high degree of sequence identity to the HA crystallized in PDB 1RVX (*Gamblin et al., 2004*). It is this HA structure that is shown *Figure 10*. The relative solvent accessibilities (RSA) values in *Figure 5* and *Figure 10* were calculated by first determining the absolute solvent accessibilities of the residues in the full trimeric HA in PDB 1RVX with the DSSP (*Joosten et al., 2011*) webserver at http://www.cmbi.ru.nl/hsspsoap/, and then normalizing by the maximum solvent accessibilities given by *Tien et al. (2013)*.

## Classification of antigenic sites and conserved receptor-binding residues

Several sub-classifications of HA residues were performed.

Conserved receptor-binding sites were any residues listed in the first table of *Martin et al. (1998)* that are also conserved in at least 90% of H1 HAs. These residues are listed in *Table 1*.

All receptor-binding residues were any residues with any atom within 5 Å of the substrate in PDB 1RVX (*Gamblin et al., 2004*). No constraint is placed on whether or not these residues are conserved in natural sequences. The residues that fall into this classification are (in sequential numbering of the WSN HA): 108, 147, 148, 149, 150, 151, 158, 166, 168, 196, 198, 199, 203, 207, 238, 239, and 241.

The *Caton et al. (1982)* antigenic-site residues are classified based on antigenic mapping of the A/Puerto Rico/8/1934 (H1N1) HA. Specifically, these are any residues listed in the third table of *Caton et al. (1982)* with the following exceptions: residue 182 (H3 numbering) is not considered for the reason explained on page 421 of *Caton et al. (1982)*, residue 273 (H3 numbering) is not considered for the reason explained on page 422 of *Caton et al. (1982)*, and residue 129 (H3 numbering) is not considered because it has gained a glycosylation site in the WSN HA that is not present in the A/Puerto Rico/8/1934 (H1N1) HA and mutation of this WSN glycosylation site can strongly affect viral growth (*Deom et al., 1986*). Overall, this gives the following set of antigenic residues, listed by sequential numbering of the WSN HA with the H3 number in parentheses: 171 (158), 173 (160), 175 (162), 176 (163), 178 (165), 179 (166), 180 (167), 169 (156), 172 (159), 205 (192), 206 (193), 209 (196), 211 (198), 182 (169), 186 (173), 220 (207), 253 (240), 153 (140), 156 (143), 158 (145), 237 (224), 238 (225), 87 (78), 88 (79), 90 (81), 91 (82), 92 (83), and 135 (122).

A second classification is done that includes the *Caton et al. (1982)* *plus* any surface-exposed residues that are in contact with these residues, using an $\alpha$-carbon to $\alpha$-carbon distance of $\leq 6.0$Å as the threshold for being in contact and classifying residues are solvent-exposed if they have an RSA of at least 20%. The rationale for this second classification is that the mapping by *Caton et al. (1982)* may have been biased towards inherently variable sites, and so other surface-exposed residues that contact these sites could also be antigenic. This classification adds the following 28 residues (listed by sequential numbering of the WSN HA) to the 29 *Caton et al. (1982)* residues: 85, 86, 89, 126, 132, 136, 137, 138, 142, 148, 150, 154, 155, 157, 170, 184, 185, 187, 202, 203, 207, 210, 212, 221, 235, 236, 239, and 252.

## Acknowledgements

We thank Paul Edlefsen for assistance with the multiple linear regression. We thank Hugh Haddox for helpful comments on the manuscript.

## Additional information

### Funding

| Funder | Grant reference number | Author |
|---|---|---|
| National Institutes of Health | R01GM102198 | Jesse D Bloom |
| Kinship Foundation | Searle Scholarship | Jesse D Bloom |

The funder had no role in study design, data collection and interpretation, or the decision to submit the work for publication.

### Author contributions

BT, Conception and design, Acquisition of data, Analysis and interpretation of data, Drafting or revising the article; JDB, Conception and design, Analysis and interpretation of data, Drafting or revising the article

### Author ORCIDs

Bargavi Thyagarajan, http://orcid.org/0000-0003-3871-6410
Jesse D Bloom, http://orcid.org/0000-0003-1267-3408

## Additional files

### Supplementary files

• Supplementary file 1. The coding sequence of the WSN HA gene used in this study is provided in FASTA format.

• Supplementary file 2. An Excel file listing the oligonucleotides used for the codon mutagenesis.

• Supplementary file 3. The site-specific amino-acid preferences as computed from the averages of the three unique replicates are provided in this supplementary file in text format. This is the

file *combined_equilibriumpreferences.txt* described at http://jbloom.github.io/mapmuts/example_ WSN_HA_2014Analysis.html.

• Supplementary file 4. The alignment of human and swine HA sequences used to build the phylogenetic trees are provided in this supplementary file in FASTA format. This is the file *H1_HumanSwine_ alignment.fasta* described at http://jbloom.github.io/phyloExpCM/example_2014Analysis_Influenza_ H1_HA.html.

## Major dataset

The following dataset was generated:

| Author(s) | Year | Dataset title | Dataset ID and/or URL | Database, license, and accessibility information |
|---|---|---|---|---|
| Bargavi Thyagarajan, Jesse Bloom | 2014 | deep mutational scanning of WSN influenza hemagglutinin | http://www.ncbi.nlm. nih.gov/biosample/ SAMN02719578 | Publicly available at NCBI BioSample. |

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
