## [Decision Letter]

Thank you for sending your work entitled “The inherent mutational tolerance and antigenic evolvability of influenza hemagglutinin” for consideration at *eLife.* Your article has been favorably evaluated by Ian Baldwin (Senior editor), a Reviewing editor, and 3 reviewers, one of whom, Sarah Cobey, has agreed to reveal her identity.

The Reviewing editor and the reviewers discussed their comments before we reached this decision, and the Reviewing editor has assembled the following comments to help you prepare a revised submission.

The reviewers agree that this paper advances knowledge on an important topic in pathogen evolution: the extent to which rapidly adapting viruses like influenza have evolved particularly evolvable phenotypes. They also judge the set of experiments important and of high quality. Finally, they recognize the novelty of combining the experimental results with an evolutionary model. This combination demonstrates that the site-specific amino acid preferences inferred from the experiments describe HA evolution more parsimoniously than traditional codon substitution models.

The reviewers provide however a number of comments whose clarification and further discussion would strengthen the arguments made in the manuscript and its eventual impact. These comments primarily concern the uncertainty and assumptions over the identity of antigenic sites and the relevant RBS residues for control purposes. It was felt that both these assumptions could be made more explicit and examined further in the ways suggested below, together with other major points.

1) The conclusions from this study are dependent on knowing which HA residues are physically located in antigenic sites. The authors use the classic [12] study to classify which HA residues are located in antigenic sites. This is a reasonable approach because the WSN/1933 virus used in the current study is antigenically similar to the PR8/1934 used in the [12] study. The Caton et al. experiments defined antigenic sites by selecting viral escape mutants in the presence of saturating amounts of individual monoclonal antibodies. Based on the Caton et al. experimental approach, viruses that possess mutations at HA residues that severely compromise viral fitness are never selected-essentially, Caton et al. defined the antigenic sites by isolating mutant viruses that were capable of growing. With this in mind, it is perhaps not surprising that these same HA residues are tolerant of mutations in the current study.

When the Caton et al. escape mutants are mapped onto the 1RVX structure, there are clusters of mutations that correspond to each antigenic site. Within each cluster, there are a few HA residues that appear to be in the cluster (thus likely in the antibody binding site) but these residues never appeared in the Caton et al. viral mutant escape experiments. It has been assumed that this is the case because either (a) there was a limited number of mutant viruses screened by Caton et al. or (b) that these residues within the antigenic site were detrimental to viral fitness.

The authors should consider making an HA structure analogous to the one shown in Figure 9, with entropy color labeling for all surface residues. This type of analysis will demonstrate if tolerability is limited to those residues identified in selection assays (assays that only identify residues capable of tolerating mutations) or also found in adjacent residues that are likely located in antibody binding sites but were not identified in the original antigenic mapping studies. It might be helpful to show surface on the structure for this type of analyses.

Another partial solution is to examine antigenic sites previously defined by other means (such as epitopes defined by crystal structures).

This is not a critical flaw but should be considered either in the discussion or by completing additional analyses as described above.

2) It is not clear that the residues most essential for function must be the most conserved ones. We know that naturally occurring neutralizing antibodies can target the RBS (Whittle et al., PNAS, 2011). There is also variation in receptor type within hosts and between species.

The main text justifies using conserved RBS as a control because these sites are expected to be strongly conserved, and the RBS is defined to exclude non-conserved RBS-associated sites. One could also argue that high mutational tolerance in the RBS could be adaptive to facilitate escape in nearby epitopes;it would be really interesting if this were the case, with the control being some non-immunogenic area near the stalk. The simplest solution may be to show that the results are unchanged when the non-conserved RBS sites are included, assuming this is the case. If it is not, it would help to know which sites the conserved RBS correspond to on the HA structure. (One would hope they're deep-in-the-pocket).

3) The paper addresses the mutational tolerance and antigenic evolvability of HA. Relative to what? Are these metrics different for other influenza proteins? Other viral glycopropteins? A cursory analysis of the HA site preference data vs. NP site preference (in [9], bioRxiv, currently in review in MBE?) would further support this claim.

4) “Big data” mutagenesis scans must sacrifice something in their broad scope; for example, quantitative as opposed to binary fitness measurements, intragenic epistasis (e.g. interactions among substitutions within HA as will occur in this experimental set-up), pool drop out bottlenecks. The authors already address many of these. It would be good to acknowledge them as shortcomings or areas for follow up work, and to discuss the recent publication by Wu et al. of a similar scanning mutagenesis that does report fitness values (Wu et al., High throughput profiling of influenza A virus hemagglutinin gene at the single-nucleotide resolution, Scientific Reports 2013).

Unlike the paper by Wu et al., however, the present manuscript places the mutational data within an evolutionary framework, which is something generally missing in most such studies. This novelty of the current manuscript could be drawn out more. In addition, both this paper and [58] present a snapshot of the (mostly) single-mutation neighbors from a single point (the sequence of WSN33 HA) in the fitness landscape. Both find that viable paths occur disproportionately at antigenic sites. The fact that these rules, when extrapolated over the phylogeny, greatly improve the fit implies that epistasis is in some ways less pervasive than we think and may not be such a huge obstacle to evolution. Most substitutions appear to be concentrated among a permissive network of sites.

5) The following aspects of the experiments should be clarified/discussed: (a) The possibility of reversion between the plasmid pool and the passaged virus is not accounted for. Granted the wild type control shows that there is low background mutation, but a weakly functional mutant could revert. (b) During transfection, one would expect that there may be viruses produced from cells expressing two different HA genes. How would such chimeric viruses confound the results in a single passage experiment? (c) Similarly to #2, wouldn't there also be viruses in which the HA on the surface has a given amino acid sequence, while the packaged genomic segments code for a different one? Multiple passages would ferret these out. (d) How were HA molecules sheared and why are there biases in sequence fragmentation? (e) Are more exposed HA residues generally more tolerant to mutations? The authors address this by correcting for relative solvent accessibility, but this description is difficult to follow. The importance of relevant solvent accessibility (RSA) is shown in Table 3 and Figure 9, but the RSA result is not stated clearly in the main text. Can the authors simplify or clarify this part?

6) The conclusions would be further strengthened if there was some way to “predict” retrospectively where HA would go next based on the model. It would be interesting to discuss this possibility along the lines of what Luksza and Lassig published in Nature this year (Nature 507:57-61).

---

## [Author Response]

*The reviewers provide however a number of comments whose clarification and further discussion would strengthen the arguments made in the manuscript and its eventual impact. These comments primarily concern the uncertainty and assumptions over the identity of antigenic sites and the relevant RBS residues for control purposes. It was felt that both these assumptions could be made more explicit and examined further in the ways suggested below, together with other major points*.

Thank you to the editors and reviewers for their very careful review of our manuscript. The critiques were very helpful, and we believe that the revisions that we have made in response have substantially improved the manuscript. Specifically:

We have included a second expanded definition of antigenic sites to show that the essential finding that HA has high antigenic evolvability is robust to the method used to classify the antigenic sites (see Response 2 below).

We have clarified that not all receptor-binding site residues have low inherent mutational tolerance – this is only true of conserved residues deep in the pocket. We now include separate classifications for conserved and all receptor-binding residues (see Response 3 below).

The suggestion to include a comparison to other viral proteins is very helpful; as suggested by the reviewers, we now compare to NP to show that high antigenic evolvability is a property of HA but not NP (see Response 4 below).

As suggested by the reviewers, we have included a comparison to the results of the Wu *et al.* study that was published while our study was in review. This comparison shows that the results of the studies are correlated, but also explains why our study provides substantially more information, and also comments on issues related to these types of high-throughput studies (see Response 5 below).

*1) The conclusions from this study are dependent on knowing which HA residues are physically located in antigenic sites. The authors use the classic*
[12]
*study to classify which HA residues are located in antigenic sites. This is a reasonable approach because the WSN/1933 virus used in the current study is antigenically similar to the PR8/1934 used in the*
[12]
*study. The Caton et al. experiments defined antigenic sites by selecting viral escape mutants in the presence of saturating amounts of individual monoclonal antibodies. Based on the Caton et al. experimental approach, viruses that possess mutations at HA residues that severely compromise viral fitness are never selected-essentially, Caton et al. defined the antigenic sites by isolating mutant viruses that were capable of growing. With this in mind, it is perhaps not surprising that these same HA residues are tolerant of mutations in the current study*.

*When the Caton et al. escape mutants are mapped onto the 1RVX structure, there are clusters of mutations that correspond to each antigenic site. Within each cluster, there are a few HA residues that appear to be in the cluster (thus likely in the antibody binding site) but these residues never appeared in the Caton et al. viral mutant escape experiments. It has been assumed that this is the case because either (a) there was a limited number of mutant viruses screened by Caton et al. or (b) that these residues within the antigenic site were detrimental to viral fitness*.

*The authors should consider making an HA structure analogous to the one shown in*
Figure 9*, with entropy color labeling for all surface residues. This type of analysis will demonstrate if tolerability is limited to those residues identified in selection assays (assays that only identify residues capable of tolerating mutations) or also found in adjacent residues that are likely located in antibody binding sites but were not identified in the original antigenic mapping studies. It might be helpful to show surface on the structure for this type of analyses*.

*Another partial solution is to examine antigenic sites previously defined by other means (such as epitopes defined by crystal structures)*.

*This is not a critical flaw but should be considered either in the discussion or by completing additional analyses as described above*.

This is a very astute comment. Essentially, the reviewers are noting that Caton *et al.* defined the antigenic sites largely by mapping monoclonal-antibody escape mutants. The Caton *et al.* study therefore may have had a bias towards mapping sites that were tolerant to mutations, potentially confounding our interpretation.

To address this concern, we have added a second expanded classification of antigenic sites which includes the specific antigenic sites mapped by Caton *et al.* plus all surface-exposed residues that are in contact with these antigenic sites. This second classification should alleviate the concern about any bias in the Caton *et al.* mapping, because even if only some residues in a cluster had mutations selected by Caton *et al.*, our expanded structural classification will also include all other residues in the same physical cluster.

In the revised manuscript, we now perform the analysis for both the original Caton *et al.* classification and this second expanded classification. The results are described in Figure 10 and Table 4. These results show that the antigenic sites have significantly higher inherent mutational tolerance regardless of whether we use the original Caton *et al.* classification or the new expanded structural classification.

In addition, we have followed the reviewers’ recommendation and included an entropy-colored surface-rendered structure of HA in Figure 10 as well as individual structures showing just the antigenic sites. This new image is panel A of the revised Figure 10. We think this image visually supports the statistical analyses showing that the expanded classification of antigenic sites has higher inherent mutational tolerance.

Finally, we have substantially revised the text in the section entitled “The inherent evolvability of antigenic sites on HA” to explain the point made by this reviewer comment and how we have addressed it. We refer the reviewers to the revised manuscript for the full text, but here is the most relevant portion:

One possible concern is that [12] mapped antigenic sites largely by selecting monoclonal-antibody escape mutants, and so these sites might be biased towards being more mutationally tolerant. We therefore also made a broader classification of ‘antigenic sites and contacting residues’ consisting of the [12] antigenic sites plus all surface-exposed residues in contact with these sites (see Methods for details). This broader classification includes all residues in regions of the HA surface targeted by antibodies, and so should not be biased by whether sites are amenable to the selection of monoclonal-antibody escape mutants. We hypothesized that both sets of anti-genic sites would have unusually high mutational tolerance.

*2) It is not clear that the residues most essential for function must be the most conserved ones. We know that naturally occurring neutralizing antibodies can target the RBS (Whittle et al.,*
*PNAS, 2011**). There is also variation in receptor type within hosts and between species*.

*The main text justifies using conserved RBS as a control because these sites are expected to be strongly conserved, and the RBS is defined to exclude non-conserved RBS-associated sites. One could also argue that high mutational tolerance in the RBS could be adaptive to facilitate escape in nearby epitopes; it would be really interesting if this were the case, with the control being some non-immunogenic area near the stalk. The simplest solution may be to show that the results are unchanged when the non-conserved RBS sites are included, assuming this is the case. If it is not, it would help to know which sites the conserved RBS correspond to on the HA structure. (One would hope they're deep-in-the-pocket)*.

This is another very astute comment. In the original version, we chose conserved receptor-binding residues as a control group for the antigenic sites because we wanted residues that we expected to be under strong constraint. However, as the reviewers point out, this could give the incorrect impression that all receptor-binding residues are highly con-served. In fact, a handful of “deep in the pocket” residues are highly conserved, but many other residues on the periphery of the receptor binding pocket are relatively amenable to change.

To address this comment, we now define two groups of receptor binding residues: the conserved ones (mostly deep in the pocket) and the set of all residues that contact the receptor. In the revised Figure 10 and Table 4 we analyze both of these groups separately, and discuss the results in the revised section “The inherent evolvability of antigenic sites on HA.” We refer the reviewers to the revised section and figure/table for full details, here we summarize the main findings by quoting from the most relevant portions of the revised section:

“For comparison to the antigenic sites, we used two classifications of receptor-binding residues (see Methods). The first classification consists of residues that have important roles in receptor binding (38) and are conserved in H1 HAs; these residues are mostly deep in the binding pocket. The second classification consists of all residues that contact the sialic-acid receptor in the crystal structure, regardless of their level of conservation. We hypothesized that the core set of conserved receptor-binding residues would have unusually low mutational tolerance, but that the set of all receptor-binding residues would have typical levels of mutational tolerance since influenza routinely escapes from antibodies that target the periphery of the receptor-binding pocket (31)...”

We then describe results that validate this hypothesis, and conclude:

“These results also show that while a core group of conserved residues deep in the receptor-binding pocket have unusually low mutational tolerance, the bulk of residues that contact the receptor are not under exceptional constraint. This fact probably explains why HA is able to escape from antibodies targeting the periphery of the receptor-binding pocket (31), and why only rare antibodies that penetrate deep into this pocket are broadly neutralizing (55).”

*3) The paper addresses the mutational tolerance and antigenic evolvability of HA. Relative to what? Are these metrics different for other influenza proteins? Other viral glycopropteins? A cursory analysis of the HA site preference data vs. NP site preference (in*
[9]*, bioRxiv, currently in review in MBE?) would further support this claim*.

This was an extremely helpful suggestion, and we believe that the changes that we have made in response have substantially strengthened the manuscript. Specifically, we now compare our results for HA to our previous results for influenza NP (which are now published in Molecular Biology and Evolution) in a new Figure 11 and Table 5. We also include a new section in the Results (“HA’s antigenic evolvability is not shared by all influenza proteins”).

*4) “Big data” mutagenesis scans must sacrifice something in their broad scope; for example, quantitative as opposed to binary fitness measurements, intragenic epistasis (e.g. interactions among substitutions within HA as will occur in this experimental set-up), pool drop out bottlenecks. The authors already address many of these. It would be good to acknowledge them as shortcomings or areas for follow up work, and to discuss the recent publication by Wu et al. of a similar scanning mutagenesis that does report fitness values (Wu et al., High throughput profiling of influenza A virus hemagglutinin gene at the single-nucleotide resolution,*
*Scientific Reports 2013**)*.

*Unlike the paper by Wu et al., however, the present manuscript places the mutational data within an evolutionary framework, which is something generally missing in most such studies. This novelty of the current manuscript could be drawn out more. In addition, both this paper and*
[58]
*present a snapshot of the (mostly) single-mutation neighbors from a single point (the sequence of WSN33 HA) in the fitness landscape. Both find that viable paths occur disproportionately at antigenic sites. The fact that these rules, when extrapolated over the phylogeny, greatly improve the fit implies that epistasis is in some ways less pervasive than we think and may not be such a huge obstacle to evolution. Most substitutions appear to be concentrated among a permissive network of sites*.

These are all very good comments. We have tried to be as transparent as possible about the shortcomings as well as the strengths of the high-throughput approach that we have used. For instance, we dedicate an entire figure (Figure 6) to transparently showing the correlations between our replicates, and honestly discuss how this figure shows that although the experiment is definitely extracting reproducibly meaningful information, it is also subject to a large amount of noise as judged by the variation among replicates. We discuss how bottlenecks and other experimental problems could contribute to this.

As the reviewers point out, while our manuscript was under review, Wu *et al.* reported a somewhat similar high-throughput study of influenza HA. The study by Wu *et al.* is clearly very valuable. However, we believe that our study contains some advantages: for instance, we survey all amino-acid mutations while Wu *et al.* survey only about 20% of them, we provide full raw data for experimental replicates and controls for possible sources of error while Wu *et al.* do not, and we perform substantially more quantitative analyses of our data including in connection to natural evolution.

To help address the points that the reviewers have made, we now include a new figure (Figure 7) and a new section in the Results that compares our study with that of Wu *et al*. We also use this new section to highlight some of the limitations of “big data” studies that the reviewers noted in their comments. Below we quote the full text of this new section, and put in bold the parts that specifically address the limitations noted by the reviewers.

As our paper was under review, [58] published the results of using a similar strategy to examine the effects of mutations to the WSN HA. In their study, the HA gene was mutated at the nucleotide level, so their experiments surveyed only amino-acid mutations accessible by single-nucleotide codon changes. As a result, they provide data on the effects of only about 20% of the 19 564 = 10716 amino-acid mutations examined in our study. Despite this limitation, their study provides a large dataset of mutational effects to which we can compare our results.

Figure 7 compares the mutational effects determined in our study to those from [58]. There is a highly significant correlation between the results of the two studies; but the inferred mutational effects are certainly not identical. Because [58] do not provide the data for replicates of their experiment, we are unable to assess whether the variability between the two different studies exceeds the variability between experimental replicates within each study. So one can imagine both biologically interesting and uninteresting explanations for the imperfect correlation between the results of the two studies. The interesting explanation is that differences in experimental methodology could lead to different selection pressures on specific mutations: for instance, [58] use A549 cells while we use MDCK-SIAT1 cells, and perhaps the impact of certain mutations is dependent on the cell line. The uninteresting explanation is that the imperfect correlation is simply due to noise in the experimental measurements. Unfortunately, it is not straightforward to distinguish between these two explanations. This difficulty in pinpointing reasons for inter-study variation highlights a limitation of the high-throughput experimental methodology employed by ourselves and [58]: while such experiments provide a wealth of data, numerous factors can create noise in these data (sequencing errors, population bottlenecks, epistasis among mutations, etc.). Realizing the full potential of such studies will therefore require extensive experimental controls and biological replicates to quantify errors and noise to enable comparisons across data sets.

Nonetheless, Figure 7 shows that there is a highly significant correlation between the results of these two high-throughput studies, despite differences in experimental methodology and unquantified sources of experimental noise. This fact suggests that both studies capture fundamental constraints on HA’s mutational tolerance. In the remaining sections, we apply the more comprehensive data generated by our study to address questions about HA’s natural evolution and antigenic evolvability.

However, beyond the text quoted above and the new Figure 7, we are unable to analyze the data of Wu et al. to with respect to the other questions about inherent mutational tolerance and viral evolution that we address in our manuscript. The reason is that Wu et al. only measure the fitness effects of an average of 4 of the possible 19 amino-acid mutations at each site, and this incomplete information is not sufficient to either assess inherent mutational tolerance or construct phylogenetic evolutionary models.

*5) The following aspects of the experiments should be clarified/discussed: (a) The possibility of reversion between the plasmid pool and the passaged virus is not accounted for. Granted the wild type control shows that there is low background mutation, but a weakly functional mutant could revert. (b) During transfection, one would expect that there may be viruses produced from cells expressing two different HA genes. How would such chimeric viruses confound the results in a single passage experiment? (c) Similarly to #2, wouldn't there also be viruses in which the HA on the surface has a given amino acid sequence, while the packaged genomic segments code for a different one? Multiple passages would ferret these out. (d) How were HA molecules sheared and why are there biases in sequence fragmentation? (e) Are more exposed HA residues generally more tolerant to mutations? The authors address this by correcting for relative solvent accessibility, but this description is difficult to follow. The importance of relevant solvent accessibility (RSA) is shown in*
Table 3
*and*
Figure 9*, but the RSA result is not stated clearly in the main text. Can the authors simplify or clarify this part*?

The reviewers ask in point (a) above about the possibility of reversions. We suspect that reversions are very rare, since the virus is only passaged once which should give a relatively small opportunity for such mutations to arise. But we acknowledge that none of our controls directly measure the rate of reversions. Our sequencing of the wildtype virus shows that mutations overall are rare, but it is possible that reversions are more common. However, a key advantage of our approach is that reversions would only affect a small fraction of our mutations – because we introduce mutations at the codon level, most (54 of 63) of the possible mutations involve multiple nucleotide changes. We deem it extremely unlikely that a single codon would revert two nucleotides during a single viral passage. To address this issue, we have revised the text.

The reviewers ask in points (b) and (c) above about chimeric viruses that express multiple HA proteins, and about viruses that lack a genotype-to-phenotype link because they possess different HA genes and proteins. We agree that this is a potential concern – however, the problem should be fixed during the low multiplicity of infection (MOI) passaging of the virus. In the current manuscript, we only perform one such passage – so the reviewers ask if it might not be desirable to also perform multiple passages to make sure the genotype-phenotype link is strong. This is a great suggestion, but in fact, in our previous work with NP (9) we did just that. Specifically, in that work, we sequenced NP after both one and two passages, and found that the results after just one passage were essentially indistinguishable from the results after two passages (see the fifth figure of http://mbe.oxfordjournals.org/content/early/2014/06/12/molbev.msu173). One reason that a single low-MOI passage is su?cient may be that there is probably substantial selection that also occurs during viral rescue in co-cultures prior to the first low-MOI passage. In any case, based on our previous NP work, we judge that one low-MOI passage is sufficient to ensure a strong genotype-phenotype link for influenza genes, and so for this reason we only performed one low-MOI passage in the current work (there are good reasons for wanting to keep the number of passages as low as possible, primarily to avoid problems like the reversion one mentioned in the reviewers’ previous comment).

To clarify this point, we have revised the manuscript (section starting “The mutant viruses generated for each replicate were passaged at a relatively low multiplicity of infection”).

The reviewers ask in point (d) how the HA molecules were fragmented. The fragmentation was done using the Illumina Nextera kit, which is widely used in the preparation of Illumina sequencing libraries. The fragmentation bias almost certainly arises due to weak sequence preferences of the transposon that mediates the fragmentation in the Nextera kit. It is known that this transposon has weak biases for certain sequences; for instance, see http://genomebiology.com/2010/11/12/r119. However, as should be clear from Figure 3—figure supplement 3 these biases are fairly mild, and lead to at most a five-fold variation in coverage at different sites.

To clarify this issue, we have revised the manuscript (section starting “In order to reduce the sequencing error rate”).

The reviewers ask in point (e) whether residues with higher relative solvent accessibility (RSA) are generally more tolerant to mutations. The answer is yes as shown in Table 4 and Figure 10, but as the reviewers point out this result was not clearly stated in the main text. We have revised the text to make this point more clearly.

*6) The conclusions would be further strengthened if there was some way to ’predict‘ retrospectively where HA would go next based on the model. It would be interesting to discuss this possibility along the lines of what Luksza and Lassig published in Nature this year (Nature 507:57-61)*.

The reviewers ask if it might be possible to predict the evolution of influenza using data from our deep mutational scanning. In particular, they mention the work of Luksza and Lassig (2014), who develop a fitness model that enables improved forecasting about which of a variety of closely related existing strains of epidemic influenza are likely to predominate in future years. We do see some ways in which our data might synergize with their approach. For instance, a key component of their model (see Equation 2 of their paper) is assigning a “mutational load” to non-antigenic mutations. One could imagine that deep mutational scanning might enable more accurate assignment of the “mutational load” caused by specific mutations, although it remains unproven whether this would actually work.

For now, we mention this possibility with the following sentence in the Discussion:

“It also may be possible to utilize high-throughput experimental data on mutational effects to better estimate the fitness of naturally occurring strains in a way that aids in prediction of the year-to-year strain dynamics of influenza (Luksza and Lassig, 2014).”

We have also provided links in the manuscript to our complete raw data and source code as well as detailed documentation of our computational analysis pipeline. We hope that this transparent availability of all our raw data will help facilitate the work of others who wish to develop approaches for viral forecasting. However, we are currently unable to offer any sort of effective predictive model ourselves, and believe that prediction of viral evolution remains a daunting problem.